# REWARD-ROBUST RLHF IN LLMS

## ABSTRACT

As Large Language Models continue to advance, Reinforcement Learning from Human Feedback is increasingly regarded as a promising approach for enhancing their capabilities and achieving more sophisticated forms of intelligence. However, the reliance on reward-model-based alignment methods introduces significant challenges due to the inherent instability and imperfections of Reward Models (RMs), which can lead to critical issues such as reward hacking and misalignment with human intentions. In this paper, we introduce a reward-robust RLHF framework aimed at addressing these fundamental challenges, paving the way for more reliable and resilient learning in LLMs. Our approach introduces a novel optimization objective that carefully balances performance and robustness by incorporating Bayesian Reward Model Ensembles to model the uncertainty set of reward functions. This allows the framework to integrate both nominal performance and minimum reward signals, ensuring more stable learning even with imperfect RMs. Empirical results demonstrate that our framework consistently outperforms baselines across diverse benchmarks, showing improved accuracy and long-term stability. We also provide a theoretical analysis, demonstrating that reward-robust RLHF approaches the stability of constant reward settings, which proves to be acceptable even in a stochastic-case analysis. Together, these contributions highlight the framework's potential to enhance both the performance and stability of LLM alignment.

## 1 INTRODUCTION

Reinforcement Learning (RL), particularly in the form of Reinforcement Learning from Human Feedback (RLHF), has become a pivotal methodology for aligning foundational models with human values and preferences. It has played a crucial role in enhancing the capabilities of Large Language Models (LLMs) to generate responses that are more helpful, harmless, and honest, contributing to significant breakthroughs such as OpenAI's o1 model (OpenAI, 2024). The standard RLHF framework consists of two key phases. First, a Reward Model (RM) is trained on preference data annotated by Human or Artificial Intelligence (AI) feedback.Following this, Proximal Policy Optimization (PPO) (Schulman et al., 2017) is applied to refine the model's performance based on the learned reward function. This structured approach ensures that LLMs operate in a manner that is consistent with ethical guidelines and user expectations, thereby enhancing their capability and trustworthiness in practical applications.

The quality of the RM is crucial to the success of PPO. A poor RM may provide incorrect signals for certain data points during the PPO training phase, ultimately compromising the performance of the fine-tuned model. Several issues arise from an imperfect RM. One such issue is *reward hacking*, where the model exploits flaws in the reward function, optimizing for behaviors that maximize the reward signal without genuinely improving task performance. Another challenge is *overfitting and underfitting*: an overfitted RM captures noise or specific patterns in the training data that fail to generalize to new data, while an underfitted model may miss important patterns altogether (Gao et al., 2023). Additionally, *misalignment with human preferences* can occur, as biases among annotators—whether human or AI (Bai et al., 2022a; Lee et al., 2023)—make it difficult to align the RM with the diverse preferences of humanity, leading to discrepancies between the model's behavior and human expectations. All the issues mentioned brings us to a critical question:

*Given that the RM is imperfect, how can we perform RM-based RLHF better?*

In this paper, we propose a reward-robust RLHF framework to overcome this challenge. To enhance the model's robustness to the reward signal while avoiding an overly conservative optimization process, we introduce a novel objective that strikes a balance between performance and robustness. The performance component is guided by a nominal RM, which serves as an approximation of the ideal reward function. Meanwhile, the robustness component accounts for worst-case scenarios by considering the uncertainty in the reward functions. To capture both the nominal reward and the uncertainty, we introduce Bayesian Reward Model Ensembles (BRME). BRME utilizes a multi-head RM, where each head outputs the mean and standard deviation (std) of a Gaussian distribution, from which the final reward is sampled. BRME has two main advantages over traditional RM that generates a single scalar as the reward: First, we demonstrate that std can reflect the confidence of each head in its output reward, allowing the output with the lowest std to be reasonably selected as the nominal reward. Additionally, we show that BRME outperforms traditional RMs in both the coverage of the reward distribution and accuracy on preference test sets. Ablation study on the RM training supports the claims above. The proposed reward-robust RLHF framework consistently outperforms standard RLHF across several widely-used benchmarks. In long run training processes, the proposed method shows stronger stability and better performance compared with the baselines.

Beyond presenting empirical performance results, we also aim to provide deeper theoretical insights. First, we delve deeper into the inherent imperfections of RMs, arguing that even with an ideal annotator, a perfect reward function is unlikely, and the resulting RM is inherently insufficient. We design a synthetic toy model that directly illustrates these limitations. Second, we provide a theoretical justification for the superiority of our method over standard RLHF. Given that the actual reward is inherently biased—resulting in either over-scoring or under-scoring during training—we show that, in the long run, under-scoring is preferable to over-scoring. Third, we conduct a stochastic case analysis within our reward-robust RLHF framework and prove that the resulting policy remains acceptable, as stochastic scenarios often arise when the RM handles out-of-distribution (OOD) data. Our analysis shows that the robustness regularization term narrows the reward distribution, making the training process closer to the constant reward setting thus more stable, which is preferable to the uncontrollable optimization driven by badly assigned rewards.

**Contributions.** 1) We propose a reward-robust RLHF framework, introducing BRME to model both nominal rewards and uncertainty, outperforming traditional RMs in reward distribution coverage and accuracy. 2) We provide theoretical insights into RM imperfections, showing through a synthetic toy model that even with an ideal annotator, a perfect reward function is unattainable. 3) We show that under-scoring is preferable to over-scoring in long-term training, given the inherent bias in accessible rewards. Additionally, we conduct a stochastic case analysis, demonstrating that the proposed framework remains effective in OOD scenarios, with the robustness regularization term stabilizing the training process by narrowing the reward distribution.

## 2 RELATED WORKS

### 2.1 REWARD-MODEL-BASED ALIGNMENT IN LLMS

The core idea behind RM-based alignment in is to use a RM, typically trained on human/AI-annotated data, to guide the optimization of the language model's policy (Christiano et al., 2017; Ouyang et al., 2022). Although RM-free algorithms such as DPO (Rafailov et al., 2024) and IPO (Azar et al., 2024) have also been developed, it is generally observed that these methods tend to be less effective compared to RM-based approaches in many scenarios (Xu et al., 2024; Yan et al., 2024b). While RM-based approach has shown success in several domains, it is not without challenges. One major issue is the potential misalignment between the RM's output and the true human preferences, which can lead to unintended behaviors, commonly known as reward hacking (Amodei et al., 2016). Additionally, the reward model's generalization ability is often limited by the quality and diversity of the annotated data, which can result in overfitting to specific data points and underfitting to others (Lee et al., 2021). These shortcomings underscore the importance of developing alternative methods that enhance reward robustness.

Recent research has explored various approaches to address these challenges. Shen et al. (2024) introduces a penalty term on the reward, named as contrastive rewards, to improve the effectiveness of RMs. Eisenstein et al. (2023) finds that RM ensembles can help mitigate reward hacking in certain scenarios. Zhang et al. (2024b) proposes a lightweight uncertainty quantification method to assess

the reliability of the reward model to avoid over-optimization. Zhang et al. (2024a); Zhai et al. (2023) uses low-rank adaptation (LoRA) to increase the diversity of RMs to improve the performance of RLHF. Yang et al. (2024b) retains the language model head and add regularization to improve the generalization capability of RMs.

Our work differs from the above research in several key aspects: 1) We propose a generally robust RLHF framework that outperforms standard RLHF across most benchmarks tested. Within this framework, we employ BRME to model an uncertainty set and introduce a trade-off objective that balances nominal performance and robustness, rather than simply using the mean or median of all the rewards as done in Eisenstein et al. (2023); Zhang et al. (2024b). In BRME, we use a Bayesian multi-head RM to characterize the uncertainty of each reward head, which distinguishes from Zhai et al. (2023). The effectiveness of BRME is supported by ablation studies. 2) Using a synthetic toy model, we demonstrate the inherent imperfections of the reward model, even when utilizing an unbiased and ideal annotator. This finding builds upon and complements prior research on annotator disagreement (Dubey et al., 2024). 3) We provide a theoretical explanation for the superiority of our method through a stochastic-case analysis. Given that the actual reward is inherently biased, tending either to over-score or under-score, we first demonstrate that, over the long term, under-scoring is preferable to over-scoring.

## 2.2 ROBUST REINFORCEMENT LEARNING

In the realm of robust RL, robustness is primarily focused on addressing uncertainties related to *transition*, *observation*, *action*, or *disturbance* (Moos et al., 2022). *Transition-robust* approaches deal with uncertainties in system dynamics by deliberately adjusting state transition probabilities (Heger, 1994; Nilim & El Ghaoui, 2005; Satia & Lave Jr, 1973; Givan et al., 2000). *Observation-robust* methods involve distorting the perceived system state to influence policy decisions (Zhang et al., 2020; Gleave et al., 2019). *Action-robust* designs modify system transitions by introducing disturbances to the agent's actions (Tessler et al., 2019). *Disturbance-robust* strategies account for external forces that introduce uncertainty into system behavior (Pinto et al., 2017).

In contrast, there has been comparatively less focus on *reward robustness*. Xu & Mannor (2006) consider an MDP with an uncertain reward function and then propose a weighted sum between a nominal and a robust performance criterion, which directly inspires our work. The trade-off can be directly made on the expected return (Xu & Mannor, 2006) or by defining a chance constraint optimization problem (Delage & Mannor, 2010). Other research targets the rectangularity assumption, which assuming that the uncertainty in each state is independent of all other states, identifying it as a primary source of the reward uncertainty (Mannor et al., 2012; 2016; Goyal & Grand-Clement, 2023). Vadori et al. (2020) proposes a risk-sensitive RL approach to handle the reward uncertainty by applying Doob decomposition on the reward. Wang et al. (2020) develops a robust RL framework where the agents can only observed perturbed rewards.

In conclusion, due to the focus of past RL applications, particularly in domains like robotics, research on reward robustness has not received sufficient attention. Notably, in the context of LLM training, the uncertainty inherent in the reward model significantly impacts final performance and hinds the progress of LLM development (Chen et al., 2024). Coste et al. (2023) introduce worst-case optimization in LLMs to mitigate overoptimization. However, their approach relies on a purely empirical method to characterize uncertainty through intra-ensemble variance, lacking both structure and interpretability. Therefore, the development of specialized reward-robust algorithms is not only necessary but also urgently required. Our work is one of the pioneering efforts to introduce reward robustness into the LLM RLHF/RLAIF pipeline.

## 3 PRELIMINARIES

**Large Language Model (LLM).** An LLM defines a $\theta$-parameterized conditional distribution $\pi_\theta(a|x)$, which takes a prompt $x$ as input and produces a response $a$. More specifically, the sampling from LLMs is performed in an auto-regressive manner:

$$\pi_\theta(a|x) = \prod_t \pi_\theta(a_t|x, a_{1:t-1}), \tag{1}$$

where $a_t$ is the $t$-th token in the response $a$ and $a_{1:t-1}$ are tokens in the response before $a_t$.

**Standard RLHF.** Training LLMs typically involves three stages: Pretraining, Supervised Fine-tuning (SFT), and RLHF. In this section, we outline the standard RLHF-PPO paradigm, widely adopted in advanced research (Ziegler et al., 2019; Ouyang et al., 2022).

Beginning with a well-trained SFT model, denoted as $\pi_0$, we proceed by sampling two responses from $\pi_0$ for each instance in a given prompt set. Subsequently, we compile a set of comparisons $\mathcal{D} = \{(x, a^+, a^-)\}$, where $a^+$ and $a^-$ denote human-preferred and human-dispreferred completions, respectively. The distribution of the preference dataset is assumed to follow the Bradley-Terry model (Bradley & Terry, 1952), i.e., the probability of response $a^+$ is better than $a^-$ is given by:

$$p_r(a^+ \succ a^- | x) = \frac{\exp(r(x, a^+))}{\exp(r(x, a^+)) + \exp(r(x, a^-))} = \sigma(r(x, a^+) - r(x, a^-)), \quad (2)$$

where $\succ$ represents the preference relation, and $\sigma(x) = \frac{1}{1+e^{-x}}$ is the sigmoid function. To train a reward model $r$, we maximize the log-likelihood of the observed preferences by minimizing the following loss function:

$$\ell_{\text{RM}}(r) = -\frac{1}{N} \sum_{(x, a^+, a^-)} \log p_r(a^+ \succ a^- | x) = - \sum_{(x, a^+, a^-)} \log \sigma(r(x, a^+) - r(x, a^-)), \quad (3)$$

where $N$ is the total number of samples in the dataset. During the RL optimization phase, we update the LLM to maximize the return from the learned reward model using the following principle:

$$\max_\theta J(\theta) = \max_\theta \frac{1}{N} \sum_x \mathbb{E}_{a \sim \pi_\theta(\cdot | x)} \left[ r(x, a) - \beta \log \frac{\pi_\theta(a|x)}{\pi_0(a|x)} \right], \quad (4)$$

where $\pi_\theta$ is initialized as $\pi_0$ and $\beta$ controls the deviation from the original model. PPO (Schulman et al., 2017) is typically used to solve the problem in practice. Algorithms that optimize the policy using a separate reward model are referred to as *RM-based* alignment.

## 4 INHERENT IMPERFECTION OF REWARD MODELS

In this section, we demonstrate that imperfection is an inherent characteristic of RMs in RLHF/RLAIF pipelines. This imperfection arises from two key factors: 1) Disagreements between annotators, which significantly affect the quality of the preference dataset. 2) The inherent difficulty in achieving an optimal reward model, even with perfectly aligned annotators.

The issue of disagreement among human annotators in the RLHF pipeline has been noted in previous research (Bai et al., 2022a). More recently, researchers have begun using AI as annotators in what is known as RLAIF, claiming comparable or even superior performance to RLHF (Lee et al., 2023; Bai et al., 2022b). However, our evaluation experiments revealed that in the RLHF process, the scoring consistency between human annotators and domain experts was approximately 70%, whereas in the RLAIF process, the consistency between multi-agent AI annotators and domain experts dropped to around 64%. The details can be found in Appendix A.

Even more surprising is the finding that, even with a perfectly aligned annotator, obtaining an optimal RM is nearly impossible. To demonstrate this, we introduce a simple toy model similar to that in Xu et al. (2024); Yan et al. (2024b). We construct discrete spaces consisting of 8 prompts and 8 responses. The LLM policy $\pi_\theta$ is modeled as a three-layer MLP that processes a one-hot vector, representing a specific response, to produce a categorical distribution over the responses. The best match for each input prompt is the response sharing the same index.

When constructing the preference dataset, assuming we have an ideal annotator, we can ideally select the perfect match for each input prompt—essentially the diagonal elements of the matrix—and two random other elements to create two preference data pairs.

The RM trained on this preference dataset and the actor model learned from the PPO process is shown in Figure 1. It is evident that both the reward model and the actor deviate significantly from the optimal. The underlying reasons can be attributed to 1) insufficient data coverage, and 2) disturbances in model training. In more complex real-world semantic spaces, these issues are amplified, further degrading the quality of the trained RM. It's important to note that in this synthetic experiment, only the coverage of the responses was considered. In actual scenarios, insufficient coverage of the prompts themselves can severely affect RM training as well.

Figure 1: Diagram and synthetic experiment results with the toy model. In the standard RLHF pipeline with the upperside gray frame, even with a dataset annotated by a global annotator, the obtained RM and the actor trained by PPO stays imperfect. In constrast, in our reward-robust RLHF pipeline with the downside orange frame, with an integration of the nominal reward functions and the uncertainty set, we can obtain the optimal actor within PPO.

## 5 REWARD-ROBUST RLHF

In this section, we formally present our reward-robust RLHF framework. According to Section 4, the golden reward function $r^*$ is not accessible. Conversely, we can only access a nominal reward function $\widehat{r}$, which is believed to be a good approximation of $r^*$ regardless of uncertainty, and a set of other reward functions $\mathcal{R}^{\text{uncertain}} = \{r_i | i = 1, 2, ..., n\}$, which can be referred to an uncertainty set.

The root of the problem in the standard RLHF pipeline lies in the excessive reliance on a single nominal reward model $\widehat{r}$ and the lack of an explicitly modeled uncertainty set, as shown in Eq. (5). Inspired by the previous work in robust RL (Xu & Mannor, 2006), we introduce a worst-case analysis to eliminate the possibility of disastrous performance and propose a robustness measurement of the policy as Eq. (6).

$$J_{\text{perform}}(\theta) := \frac{1}{N} \sum_x \mathbb{E}_{a \sim \pi_\theta(\cdot|x)} \left\{ \widehat{r}(x, a) - \beta \log \frac{\pi_\theta(a|x)}{\pi_0(a|x)} \right\}, \tag{5}$$

$$J_{\text{robust}}(\theta) := \min_{r \in \mathcal{R}^{\text{uncertain}}} \frac{1}{N} \sum_x \mathbb{E}_{a \sim \pi_\theta(\cdot|x)} \left\{ r(x, a) - \beta \log \frac{\pi_\theta(a|x)}{\pi_0(a|x)} \right\}, \tag{6}$$

$$J_\lambda(\theta) := \lambda J_{\text{perform}}(\theta) + (1 - \lambda) J_{\text{robust}}(\theta). \tag{7}$$

In contrast to relying solely on the nominal reward model or the pure worst-case analysis, we use a trade-off term between the performance and the robustness to be our objective function as Eq. (7). There are several disadvantages to use Eq. (6) as the objective directly. 1) It often leads to an overly conservative solution, resulting in mediocre performance across all situations. 2) The desirability of the solution heavily depends on the precise modeling of the uncertainty set, which is challenging in the context of LLM training. 3) If the nominal reward model $\widehat{r}$ are close to the golden one $r^*$, the performance under nominal reward signal can provide valuable insights into predicting performance under $r^*$. 4) There is an inherent trade-off between worst-case performance and nominal performance—maximizing one often comes at the expense of the other. On the other hand, by relaxing both criteria, it is possible to achieve a well-balanced solution that offers satisfactory nominal performance while maintaining reasonable robustness to reward uncertainty.

We first demonstrate the effectiveness of the method in the toy model setting. For simplicity, we use different random seeds to train three additional different RMs, forming the uncertainty set $\mathcal{R}^{\text{uncertain}}$, and set $\lambda = 0.4$. The results, illustrated in the red dotted frame in Figure 1, show that the optimal

actor in PPO is achieved, in contrast to the standard case where only the nominal reward model is used.

In the real LLM scenario, the situation becomes more complex. We will delve into the details of uncertainty set modeling, the selection of the nominal reward function, and the evaluation of BRME and end-to-end PPO performance in the following subsections. As our framework can be easily integrated into existing RLHF pipelines, we will also provide experimental results to illustrate best practices, such as how to select the trade-off hyperparameter $\lambda$.

## 5.1 BRME: UNCERTAINTY SET MODELING

In this section, we will show how we model the uncertainty set $\mathcal{R}^{\text{uncertain}}$ as well as the nominal reward function $\hat{r}$. We propose *Bayesian Reward Model Ensembles (BRME)*: We train a multi-head Bayesian reward model where the reward is modeled as a Gaussian distribution. Each head $i$ has two outputs: one representing the mean and the other representing the std. A sample from this distribution is then output as the reward.

The diagram for BRME is shown in Figure 2. To facilitate deployment and conserve computing resources, we do not train multiple RMs independently as an ensemble; instead, we use parameter sharing. All reward heads share a common base model, which serves as a feature extractor. During training, each data sample is randomly assigned to a head for training, with reparameterization employed to address the non-differentiability of the sampling process.

The training process is divided into two stages. In the first stage, we train a traditional one-head RM following the Maximum Likelihood Estimation (MLE) loss in (3). In the second stage, we leverage a Mean Squared Error (MSE) loss (8) to train the RM, which is first introduced in (Wu et al., 2024). The use of MSE loss to train the BRME ensures that: 1) the output's std reflects the confidence of the model (see Appendix B.1), 2) the scoring range of each reward head is consistent. During prediction, each sample is evaluated by all heads, and the mean output by the head with the smallest std is selected as the nominal reward. We defer the thorough training pipeline, the theoretical analysis and detailed performance evaluation of BRME to Appendix B.

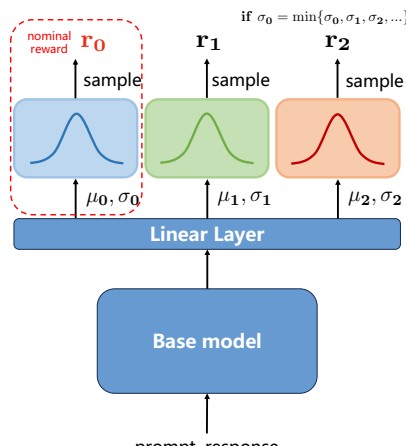

Figure 2: Diagram for BRME. Each head outputs the mean and the std of the corresponding reward distribution and reparametrization is emplyed to address the non-differentialbility.

## 5.2 EXPERIMENTAL RESULTS

### 5.2.1 EXPERIMENTAL SETUP

**Models.** We use LLaMa3-8B-Instruct (Dubey et al., 2024) as the initialization of the actor model. In BRME setting, we train a single-head RM and a 5-head BRME starting from LLaMa3-8B-Instruct (Dubey et al., 2024) as is described in Appendix B. The single-head RM is used as both the first stage model for BRME training and the initialization of the critic model. BRME is used solely as the reward signal source in PPO.

**Datasets.** For the training process, we use HH-RLHF (Bai et al., 2022a) and UltraFeedBack (Cui et al., 2024) to train the BRME. HH-RLHF, UltraFeedBack (Cui et al., 2024) , along with an internal prompt dataset collected by the PM team, is employed to implement the PPO algorithm. Details of the datasets are deferred to Table 6 in Appendix D. For performance evaluation, we select ARC (Clark et al., 2018), LAMBADA (Paperno et al., 2016), PIQA (Bisk et al., 2020), SciQ (Welbl et al., 2017), WinoGrande (Sakaguchi et al., 2019), TQA (Lin et al., 2022), MMLU (Hendrycks et al., 2020), GSM8K (Cobbe et al., 2021), FDA (Arora et al., 2023), EQ-Bench (Paech, 2023),

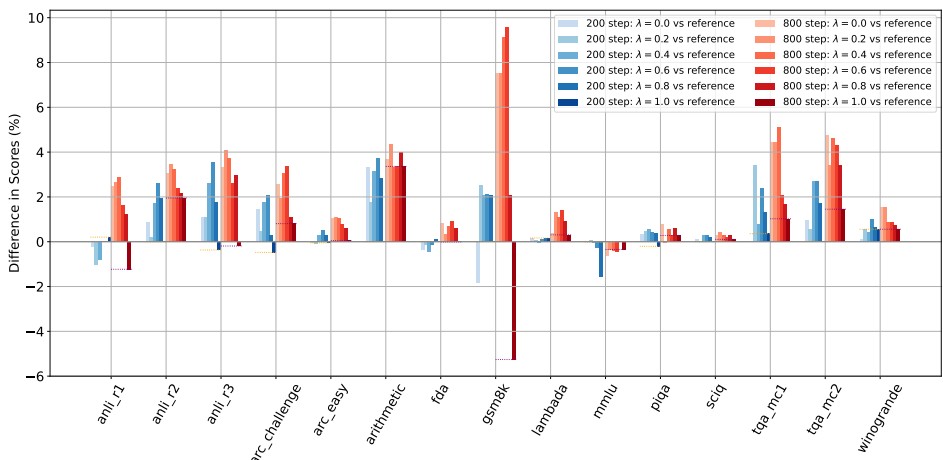

Figure 3: Evaluation results of reward-robust RLHF framework with the performance-robustness trade-off hyperparameter $\lambda$ varying, where the objective function $J_\lambda(\theta) := \lambda J_{\text{perform}}(\theta) + (1 - \lambda)J_{\text{robust}}(\theta)$. Note that when $\lambda = 1$, the algorithm reduces to standard RLHF with a single nominal reward model.

Arithmetic (Brown et al., 2020), and ANLI (Nie et al., 2020) as our benchmarks. The evaluation dimensions include robustness, general knowledge, numerical computation, emotion reading, information extraction, reasoning, context understanding, and commonsense. Detailed descriptions of the datasets are shown in Table 7 in Appendix D.

**Other details.** The RM training and PPO experiments are conducted on 24 Nvidia H800-SXM-80GB GPUs in 3 nodes using DeepSpeed library, ZeRO stage 2 (Rasley et al., 2020), and Hugging-Face Accelerate (Gugger et al., 2022). In PPO process, the actor model and the critic model occupy 10 gpus respectively. The reference model and BRME occupy 2 gpus respectively. We use AdamW optimizer (Loshchilov et al., 2017). The experience batch size in PPO is set to be 128.

### 5.2.2 MAIN RESULTS

We incrementally increase the performance trade-off hyperparameter $\lambda$ from 0 to 1 in intervals of 0.2 and repeated PPO training under each setting for 800 steps. It is important to note that when $\lambda = 1$, the algorithm degrades to standard RLHF with a single nominal RM. The performance evaluation results at 200 and 800 steps are shown in Figure 3.

In the short run (200 steps), although there are a few exceptions—such as ANLI-r1 and LAMBADA, where $\lambda = 0$ outperforms the other settings—in most cases, the trade-off versions with $\lambda = 0.4$ and $\lambda = 0.6$ show better performance. This suggests that, even early in training, incorporating a balance between performance and robustness offers notable advantages. The ability of the reward-robust RLHF framework to temper the optimization process appears to result in more stable performance gains compared to standard RLHF. Compared with standard RLHF, the average accuracy of reward-robust RLHF increases by 0.99% and 1.40% respectively when $\lambda = 0.4$ and 0.6. On certain datasets, such as arithmetic and ANLI-r3, the improvement exceeds 3%.

Over the long run (800 steps), the advantages of incorporating robustness measurement become even more pronounced. Nearly all experimental groups with $\lambda \neq 1$ exhibit improved performance by the end of 800 steps, confirming the long-term benefits of reward-robust RLHF. In contrast, standard RLHF with $\lambda = 1$ not only fails to capitalize on the full optimization process but, in some cases, even results in negative performance growth. For example, in tasks like ANLI-r1 and GSM8K, we observe a decrease in model capability during the additional 600-step optimization process under standard RLHF. This highlights a key limitation of the purely performance-driven RLHF approach: its vulnerability to the risk of overfitting or misguidance from imperfect reward signals. Compared with standard RLHF, the accuracy of reward-robust RLHF increases by 2.42% and 2.03% respectively when $\lambda = 0.4$ and 0.6.

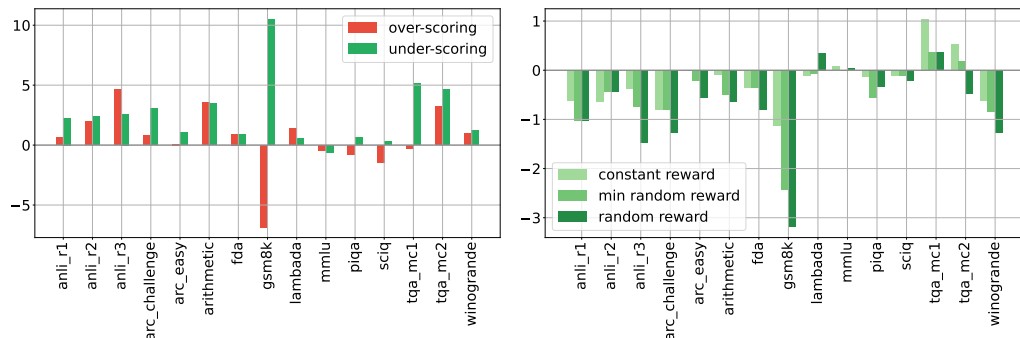

(a) Over-scoring vs. under-scoring. In most cases, under-scoring outperforms over-scoring.

(b) Stochastic case analysis. Constant rewards had the smallest decline, random rewards the largest, with minimum random rewards in between.

Figure 4: Performance shift in Section 6.1 and 6.2 suggest: 1) under-scoring is generally preferable to over-scoring, and 2) leveraging the minimum reward in the uncertainty set helps mitigate performance decline when the RM underperforms.

On tasks where the RM performs notably poorly, such as MMLU, we notice that PPO training under standard RLHF can degrade model performance, evidenced by a drop in accuracy, which is reflected in the bar graphs pointing downward. This degradation likely results from the model being optimized based on unreliable or misleading reward signals, which causes the policy to drift away from optimal behavior. In contrast, the reward-robust RLHF approach, which balances performance with robustness ($\lambda = 0.4$ or $\lambda = 0.6$), mitigates this decline. By narrowing the optimization focus and stabilizing the reward signal, the model is better able to resist overfitting to faulty rewards, ultimately preserving and improving performance. The results suggest that in settings where training data is sparse and the RM is significantly imperfect, adopting a reward-robust approach can effectively stabilize training and prevent further performance degradation over time.

## 6 DISCUSSION

In this section, we provide insights into why the proposed reward-robust RLHF framework is effective and how it improves upon the previous standard pipeline. Given that the reward signal is inherently imperfect, we empirically demonstrate that over-scoring is more detrimental than under-scoring, which supports our choice of minimum return as a robustness measure. Additionally, we show that in the stochastic case where rewards are given randomly, selecting the minimum reward ensures that the trained model in PPO remains at least acceptable. In Section 6.3, we also do ablation study to compare our method with traditional RM with MLE loss as well as other integration strategies such as using mean reward.

### 6.1 OVER-SCORING VS. UNDER-SCORING

We have demonstrated that RMs are inherently imperfect (Section 4), meaning the rewards used in the PPO training are either over-scored or under-scored. Through PPO training on minimum reward and maximum reward, we show that over-scoring is significantly more harmful than under-scoring in PPO training. The results of this comparison are illustrated in Figure 4a. In 12 of the total 16 benchmarks, under-scoring setting outperforms the over-scoring setting (see Table 4, Appendix C.2).

We also specifically analyzed the dynamic performance changes under two different settings. We conducted PPO training for 1000 steps, measuring performance every 200 steps. Generally, training with the minimum reward setting proved to be more stable: in the early stages, in some tasks such as ARC and PIQA, the minimum reward setting showed slower progress compared to the maximum reward setting. However, as training continued, the minimum reward generally led to a more consistent and stable improvement, whereas the maximum reward setting gradually resulted in performance degradation. A comprehensive result on all datasets is provided in Figure 5a. Additional results and discussions are deferred to Appendix C.

In our reward-robust RLHF framework, rewards tend to be under-scoring. To see this, we test BRME on UltraFeedBack (Cui et al., 2024), diverse preference dataset with annotated rewards. We approximately treat the annotation as the ground-truth reward, measuring the scoring gap of each reward head as well as the minimum reward head and the ground-truth reward. Note that the rewards are normalized with respect to its resource. The result is shown in Figure 5b. For a single head, the number of under-scoring and over-scoring cases is roughly equal. However, in the reward-robust RLHF setting, selecting the minimum rewards leads to under-scoring becoming more frequent.

We propose the following hypotheses to explain these observations: 1) In reward-robust RLHF, the optimization process tends to be more conservative, which could benefit long-term exploration. In essence, reward-robust RLHF might prioritize optimizing the lower bound of the return function. 2) In language tasks, exploring an incorrect direction may be more detrimental than rejecting a correct one. For a given prompt, there are often multiple valid responses. Even if the model misses one correct optimization path, it may still be able to explore alternative directions. However, with a more aggressive optimization strategy (such as over-scoring), the model might be more prone to pursuing a wrong direction, potentially resulting in reward hacking.

## 6.2 STOCHASTIC CASE ANALYSIS

If the RM is poorly trained or handling OOD data, the extreme scenario is that rewards are given randomly. In this section, we demonstrate that even in such a stochastic case, the reward-robust RLHF framework still yields an acceptable model, and we provide an explanation for why this is the case. First, we have the following lemma,

**Lemma 1.** If the reward model provides a constant reward for all actions during PPO training, the actor will not be optimized, as the gradient of the PPO objective function with respect to the policy parameters will be zero.

The proof for Lemma 1 is straightforward since it is easy to find the advantage functions for all states will be all zero when the reward is a constant. We defer the complete proof to Appendix E.

If we use reward robust RLHF training (setting $\lambda = 0$ in Eq. 7), as we are choosing the minimum reward, the range for reward will be narrowed (see Figure 5b and Figure 8b) and is closer to the constant reward situation. Since the starting point of our training is usually a well-trained SFT model, even if the optimization process degrades, the model can still maintain relatively good performance. In the contrast, if we use a random reward model, the optimization will be uncontrollable and the model will collapse in a fast speed.

We also conduct PPO experiments for 200 steps with random rewards on real LLMs. The experimental setup is identical to that described in Section 5.2, except that the rewards from each head were sampled from a Gaussian distribution $\mathcal{N}(0, 1)$. A control group was also set up, in which all rewards were set to zero. The results are shown in Figure 4b. Although nearly all performances declined, the constant reward setting exhibits the smallest decrease, while the random reward setting shows the largest decline. The minimum random reward setting falls in between, indicating that it can help the model remain stable even in a highly unpredictable random-reward environment.

**Remark 1.** In the practice, the model optimization process is likely to oscillate, but it will eventually converge back to the inital-SFT model in a long run theoretically. The reason why it will oscillate is that the critic model is not ideal in the early stage, so the advantage $\widehat{A}_t$ is not 0 even all the rewards given are identical. However, as the KL divergence ($-\beta \log \frac{\pi_\theta(a|x)}{\pi_0(a|x)}$) is also considered in the optimization objective (4), the actor policy will be eventually pulled back close to the reference policy.

## 6.3 ABLATION STUDY

Our framework has two main different parts with the previous RLHF method. One is the BRME setting trained with MSE loss. In most previous works, the RM directly output a scalar to be the final reward and the loss function is MLE(3). Another is the integration strategy. We use the trade-off version between the lower-bound reward and a nominal reward, while there are other strategies such as using the mean reward Eisenstein et al. (2023). Here we provide the ablation study results.

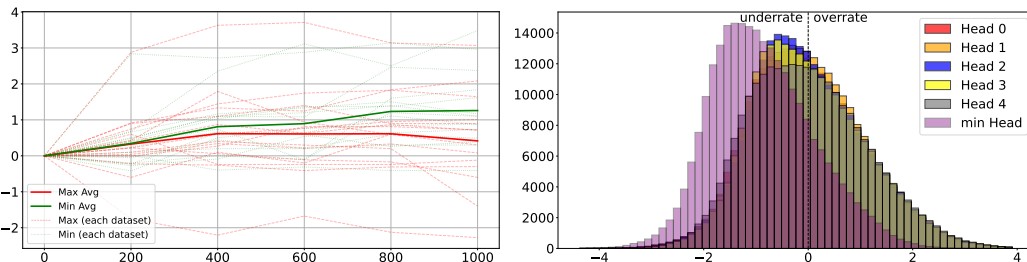

(a) Averaged performance improvement through maximum reward and minimum reward.

(b) Value range of each reward head in BRME and the minimum reward.

Figure 5: The left figure shows that as training progresses, the performance under the minimum reward setting steadily improves, indicating that conservative optimization benefits the long-term PPO optimization of LLMs. The right figure illustrates the effect of minimizing the reward on its value distribution, reducing the range and making under-scoring dominant.

**Comparison with RM trained with MLE loss.** We trained another RM using MLE loss on the same training data as mentioned in Appendix B. The resulting single-head RM was then compared with the nominal head of the BRME. To assess performance, we evaluated the accuracy on a series of preference datasets, including General QA, Writing, Comprehension, and Math (Table 5). One key advantage of BRME is its ability to effectively model the diversity within the uncertainty set, which directly influences optimization Mannor et al. (2016). To demonstrate this, we also compared the reward signal distribution coverage (Figure 7). Additionally, we conducted PPO training using both RMs to compare their effects on the actors' performance. The results consistently support BRME's superiority, with detailed findings provided in Appendix C.3.

**Comparison with other reward integration strategies.** We conducted the same PPO experiment using an average of all BRME heads and compared it with the min/max integration strategies. Additionally, we evaluated the trade-off strategy against the mean reward approach. The result is shown in Figure 9c and Figure 9b, Appendix C.4. The performance of the mean strategy falls between the max and min strategies, but in the later stages of training (after 800 steps), the mean strategy also tends to lead to a decline in model performance. On most datasets where PPO has a significant effect, the reward-robust RLHF setting with a trade-off parameter of $\lambda = 0.6$ outperforms the mean strategy.

## 7 CONCLUSIONS AND FUTURE WORKS

In this paper, we proposed a reward-robust RLHF framework to address the problem of reward hacking in LLM alignment. We demonstrated that imperfection is an inherent characteristic of current reward model training, leading to the model exploring incorrect optimization directions. To mitigate this issue, we trained BRME with multi-head outputs to model the uncertainty set of the reward function. We showed that the head with the minimum std effectively models the nominal reward function, representing the most confident scoring output. The newly proposed robustness-performance trade-off objective was proven effective, consistently outperforming baselines across most benchmarks. Furthermore, we demonstrated that under-scoring is preferable to over-scoring when dealing with imperfect RMs. Even in the stochastic-case scenario, where rewards are assigned randomly, the reward-robust RLHF framework still yields an acceptable model. Finally, we acknowledge the framework's limitations, which are discussed in detail in Appendix G.

Since our method can be easily integrated into existing pipelines, there is potential to further improve performance by incorporating additional reward sources to better model the uncertainty set. Future work will explore the adoption of heterologous reward sources, including RMs trained on diverse datasets and direct scoring from closed-source LLM APIs such as GPT-4, as well as other markers mentioned in (Yan et al., 2024a; Liu et al., 2023). The advantage of using heterologous models lies in their diverse base training datasets, which result in more varied reward scores, thereby improving the coverage of the uncertainty set. Preliminary exploration results on heterologous reward fusion are provided in Appendix F.

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

## A    ANNOTATOR DISAGREEMENT IN RLHF AND RLAIF

For human-annotated data, we conducted an agreement test on 209 data points, each containing one prompt and two responses. The prompts were selected from an internal dataset by the PM team, and the responses were generated by Baichuan2-13B (Yang et al., 2023). Data categories include general knowledge, logical reasoning, tables, mathematics, etc. We used two distinct annotator groups: one consisted of highly educated internal annotators who had undergone multiple rounds of specialized

annotation training, referred to as the *Expert Group*. The other group was composed of external annotators hired from the general public, referred to as the *External Group*. For each data point, annotators were tasked with a Good-Same-Bad evaluation: 1) G: response 1 is better than response 2. 2) S: both responses are of the same quality. 3) B: response 1 is worse than response 2. We compared the G/S/B annotations for the same data between the two groups. Across all samples, the consistency rate was 70%, with a 4.5% rate of opposite judgments (one group scored G while the other scored B). When only considering G/B samples, the consistency rate increased to 77%, with an opposite rate of 6.5%.

We also established an AI feedback pipeline, primarily using the GPT-4 API as the annotator, referred to as the *AI Group*. The PM team biasedly sampled 85 examples, focusing on cases where the *Expert Group* and *External Group* showed inconsistent labeling. For these data points, the consistency between the *Expert Group* and the *AI Group* was 64%, while the consistency between the *External Group* and the *AI Group* was 44%. In 9% of the cases, the *Expert Group* differed from both the *External Group* and the *AI Group*.

These results indicate that, whether in the RLHF or RLAIF process, annotator disagreement remains a significant challenge currently, which complicates RM training and presents obstacles that must be addressed.

## B  BAYESIAN REWARD MODEL ENSEMBLES (BRME)

In this section, we will provide the training detail, the theoretical explanation, and the empirical performance of the proposed Bayesian Reward Model Ensembles (BRME).

### B.1  TRAINING PIPELINE

The training process is divided into two stages. In the first stage, we train a normal one-head RM following the loss function in (3). In the second stage, we leverage a MSE loss to train the RM, which is first introduced in (Wu et al., 2024). The loss function for a single head $i$ is given by:

$$\ell_i = \left\{ r_i^+ - \alpha \left[ \widehat{p}(a^+ \succ a^-) - \frac{1}{2} \right] \right\}^2 + \left\{ r_i^- - \alpha \left[ \widehat{p}(a^+ \prec a^-) - \frac{1}{2} \right] \right\}^2, \tag{8}$$

where $\widehat{p}(a^+ \succ a^-)$ is derived from a separate Bradley-Terry model, as defined in Eq. (2), and is the output of the normal RM trained in the first stage. The reward is modeled as a Gaussian distribution, with each head $i$ producing two outputs: one representing the mean and the other representing the std. A sample from this distribution is then output as the reward. We employ the reparameterization handle the non-differentiability of the sampling process:

$$r_i^+ = \mu_i^+ + a \cdot \sigma_i^+, \quad r_i^- = \mu_i^- + a \cdot \sigma_i^-,$$

where $a$ is a parameter sampled from a standard Gaussian distribution $\mathcal{N}(0, 1)$.

**Why the std reflects the confidence of the head?**  The use of MSE loss to train the BRME ensures that the output's std reflects the confidence of the model. To understand this, we compute the gradient of the MSE loss with respect to the std outputs $\sigma_i^+$ and $\sigma_i^-$, where $k = \widehat{p}(a^+ \succ a^-) - \frac{1}{2}$ is a constant,

$$\nabla_{\sigma_i^+} \ell_i = 2a \cdot (\mu_i^+ + a \cdot \sigma_i^+ - \alpha \cdot k), \quad \mathbb{E}_a[\nabla_{\sigma_i^+} \ell_i] = \mathbb{E}_a[2a^2 \sigma_i^+] \geq 0,$$
$$\nabla_{\sigma_i^-} \ell_i = 2a \cdot (\mu_i^- + a \cdot \sigma_i^- - \alpha \cdot k), \quad \mathbb{E}_a[\nabla_{\sigma_i^-} \ell_i] = \mathbb{E}_a[2a^2 \sigma_i^-] \geq 0.$$

A positive gradient indicates that with more optimization steps, the output std $\sigma_i^+$ (or $\sigma_i^-$) decreases, reflecting higher confidence from the reward head in its scoring.

**Why the method is Bayesian?**  The reward generated by the first-stage reward model, represented by $\widehat{p}(a^+ \succ a^-)$, serves as an implicit prior. In the second stage, we refine this prior by using additional training data, aligning with the Bayesian perspective of updating an initial belief based on new evidence.

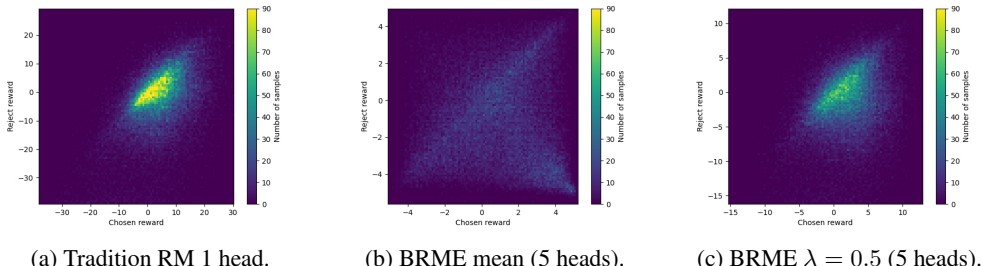

(a) Tradition RM 1 head.     (b) BRME mean (5 heads).     (c) BRME $\lambda = 0.5$ (5 heads).

Figure 7: Reward distribution coverage comparison between BMRE and traditional RM.

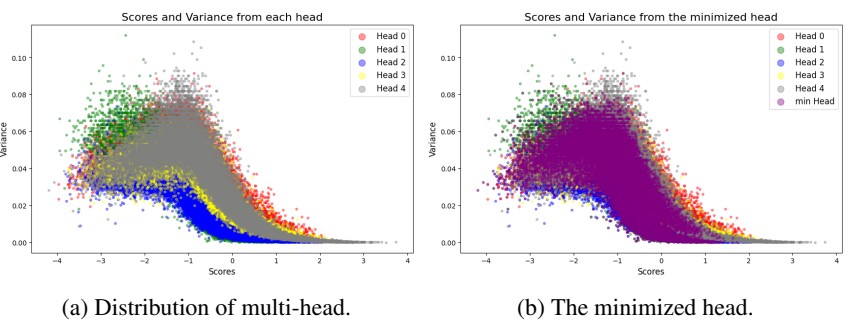

(a) Distribution of multi-head.       (b) The minimized head.

Figure 8: The over-scoring effect.

Specifically, the first stage provides point estimates through MLE, which we consider as prior beliefs. In the second stage, we model the reward as a Gaussian distribution with mean $\mu_i$ and standard deviation $\sigma_i$, incorporating uncertainty around these initial estimates. This distributional approach transforms the prior point estimates into a posterior distribution, representing the updated belief about the rewards.

## B.2   DATA PARTITION

We use different data to train each head. We first shuffled the data in Table 6 and randomly assigned each data point to one of the reward heads, with each head consuming only 20% of the total data. This approach helps distinguish the score distribution for each reward head.

## B.3   EXPERIMENTAL RESULTS OF BRME

We conduct several experiments to directly test the performance of BRME. We train another tradition RM with MLE loss to be the baseline. One of the main reasons that we use ensembling is to expand the reward coverage and obtain an informative uncertainty set. We use a separated preference testset to measure the coverage. The results are shown in Figure 7. We use the rewards of chosen response and rejected response as the horizontal and vertical axes and visualize the distribution results. BRME is measured by mean integration (Figure 7b) and trade-off integration with $\lambda = 0.5$ (Figure 7c). It can be seen that BRME rewards are more widely distributed.

Another important performance measurement is the reward margin between chosen and rejected responses. A larger reward margin indicates a greater ability of the RM to differentiate between better and worse responses,

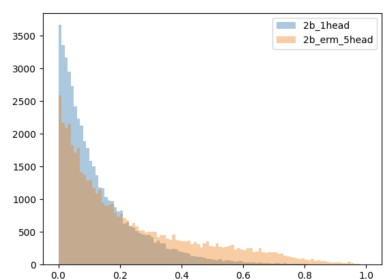

Figure 6: Normalized reward margin between chosen and rejected responses.

which is critical for guiding the optimization process effectively. We provide a reward margin comparison between BRME and the traditional RM in Figure 6. The results show that the reward margin in BRME is significantly larger than in the traditional RM, suggesting that BRME has a stronger capability to distinguish between high-quality and low-quality responses. This improved differentiation leads to more accurate guidance during training, helping the model focus on better responses more consistently. The increased reward margin also reduces the likelihood of reward hacking, as the model is less likely to be misled by small differences between responses. Overall, the larger reward margin in BRME demonstrates its advantage in promoting better alignment with human preferences and improving the robustness of the training process.

## C ADDITIONAL EXPERIMENTS

### C.1 DETAILED EXPERIMENTAL RESULTS IN SECTION 5.2

In this section, we provide the detailed experimental results in Section 5.2. Detailed data related to Figure 3 is shown in Table 1, Table 2 and Table 3. In a short run, when training step is 200, $\lambda = 0.4$ enjoys the best performance, where 9 of 16 benchmarks outperforms the baseline and the standard RLHF ($\lambda = 1.0$). However, there are still 2 benchmarks (ANLI-r1 and LAMBADA) where the standard RLHF is the best. It indicates that at the very beginning of the PPO process, choosing the nominal reward function to be the signal may lead to faster improvement.

As the training process is prolonged to 800 steps, the benefits of incorporating robustness into the RLHF framework become more evident. The results show that settings with $\lambda \neq 1.0$ generally outperform standard RLHF across a majority of the benchmarks. Notably, $\lambda = 0.2, 0.4$ and $0.6$ continue to exhibit strong performance, particularly on tasks such as ARC-challenge and GSM8K, where the reward-robust RLHF outperforms standard RLHF with a clear margin. Specifically, on GSM8K, $\lambda = 0.6$ results in a 4.93% improvement over standard RLHF. These results highlight the long-term advantages of balancing performance and robustness, as the reward-robust setting helps avoid the pitfalls of overfitting to imperfect reward signals, which is more likely to occur under standard RLHF.

Moreover, in tasks like ANLI-r1, which initially favored standard RLHF at 200 steps, the performance of $\lambda = 0.4$ surpasses the nominal reward strategy by the 800-step mark. This indicates that while the nominal RM may provide faster short-term gains, incorporating robustness into the optimization process enables more consistent long-term improvements. Similarly, on the ARC-challenge, $\lambda = 0.6$ outperforms standard RLHF with a 1.35% gain, further confirming the utility of the reward-robust approach for challenging tasks where the RM may struggle with accuracy in the initial stages.

The degradation in performance observed in some tasks, such as ANLI-r1 and GSM8K, underscores the limitations of purely performance-driven RLHF, particularly in cases where the reward signal is noisy or unreliable. Over the extended training period of 800 steps, standard RLHF tends to overfit or follow misleading reward signals, leading to a decline in model performance. This is particularly concerning in tasks like GSM8K, where standard RLHF results in negative performance growth, while reward-robust strategies maintain stability and even improve accuracy.

On the other hand, the conservative nature of the reward-robust RLHF framework, mitigates this risk. By accounting for uncertainty in the RM, the framework effectively narrows the optimization space, allowing the model to avoid over-optimization based on potentially erroneous reward signals. This results in more stable, long-term gains in performance, as evidenced by the consistent improvements across a range of tasks.

In summary, while standard RLHF may achieve faster short-term improvements in some cases, the reward-robust RLHF framework proves to be more reliable over longer training periods. The inclusion of robustness not only enhances performance but also stabilizes the training process, making it more resilient to the inherent imperfections of RMs, especially in complex and challenging tasks.

| | Baseline | $\lambda = 0.0$ | $\lambda = 0.2$ | $\lambda = 0.4$ | $\lambda = 0.6$ | $\lambda = 0.8$ | $\lambda = 1.0$ |
|---|---|---|---|---|---|---|---|
| ANLI-r1 | 0.4880 | 0.4870 | 0.4830 | 0.4840 | 0.4880 | 0.4880 | **0.4890** |
| ANLI-r2 | 0.4610 | 0.4650 | 0.4620 | 0.4690 | **0.4730** | 0.4700 | 0.4610 |
| ANLI-r3 | 0.4492 | 0.4541 | 0.4542 | 0.4608 | **0.4650** | 0.4570 | 0.4475 |
| ARC-challenge | 0.5324 | 0.5401 | 0.5350 | 0.5418 | **0.5435** | 0.5340 | 0.5299 |
| ARC-easy | 0.8165 | 0.8161 | 0.8157 | 0.8190 | **0.8207** | 0.8188 | 0.8161 |
| Arithmetic | 0.8568 | 0.8852 | 0.8718 | 0.8838 | **0.8888** | 0.8812 | 0.8568 |
| FDA | 0.7804 | 0.7776 | 0.7795 | 0.7768 | 0.7795 | **0.7812** | 0.7804 |
| GSM8K | 0.3320 | 0.3260 | **0.3404** | 0.3389 | 0.3390 | 0.3389 | 0.3321 |
| LAMBADA | 0.7180 | **0.7192** | 0.7186 | 0.7176 | 0.7188 | 0.7191 | **0.7192** |
| MMLU | 0.6381 | 0.6383 | **0.6384** | 0.6379 | 0.6365 | 0.6281 | 0.6380 |
| PIQA | 0.7878 | 0.7905 | 0.7916 | **0.7922** | 0.7911 | 0.7908 | 0.7862 |
| SciQ | 0.9640 | 0.9650 | 0.9640 | **0.9670** | **0.9670** | 0.9660 | 0.9640 |
| TQA-mc1 | 0.3610 | 0.3611 | **0.3733** | 0.3638 | 0.3696 | 0.3658 | 0.3623 |
| TQA-mc2 | 0.5165 | 0.5214 | 0.5195 | **0.5304** | **0.5304** | 0.5254 | 0.5166 |
| Winogrande | 0.7174 | 0.7182 | 0.7214 | 0.7206 | **0.7245** | 0.7219 | 0.7214 |
| EQ-Bench | 61.680 | 62.538 | 62.259 | 63.088 | **63.859** | 62.524 | 61.680 |

Table 1: Accuracy improvement for each $\lambda$ on all the benchmarks tested at 200 step in PPO. Baseline here is the SFT model, which is LLaMa3-8B-Instruct (Dubey et al., 2024).

| | Baseline | $\lambda = 0.0$ | $\lambda = 0.2$ | $\lambda = 0.4$ | $\lambda = 0.6$ | $\lambda = 0.8$ | $\lambda = 1.0$ |
|---|---|---|---|---|---|---|---|
| ANLI-r1 | 0.4880 | 0.5000 | 0.5010 | **0.5020** | 0.4960 | 0.4940 | 0.4820 |
| ANLI-r2 | 0.4610 | 0.4750 | **0.4770** | 0.4760 | 0.4720 | 0.4710 | 0.4700 |
| ANLI-r3 | 0.4492 | 0.4642 | **0.4675** | 0.4658 | 0.4608 | 0.4625 | 0.4483 |
| ARC-challenge | 0.5324 | 0.5461 | 0.5427 | 0.5486 | **0.5503** | 0.5383 | 0.5367 |
| ARC-easy | 0.8165 | 0.8249 | **0.8253** | 0.8249 | 0.8228 | 0.8215 | 0.8169 |
| Arithmetic | 0.8568 | 0.8882 | **0.8940** | 0.8852 | 0.8856 | 0.8912 | 0.8856 |
| FDA | 0.7804 | 0.7867 | 0.7831 | 0.7858 | **0.7877** | 0.7849 | 0.7803 |
| GSM8K | 0.3320 | 0.3571 | 0.3571 | 0.3624 | **0.3639** | 0.3389 | 0.3146 |
| LAMBADA | 0.7180 | 0.7206 | 0.7275 | 0.7257 | **0.7281** | 0.7246 | 0.7202 |
| MMLU | 0.6381 | 0.6359 | 0.6356 | 0.6352 | 0.6352 | **0.6382** | 0.6359 |
| PIQA | 0.7878 | **0.7938** | 0.7873 | 0.7922 | 0.7900 | 0.7927 | 0.7900 |
| SciQ | 0.9640 | 0.9670 | **0.9680** | 0.9670 | 0.9660 | 0.9670 | 0.9650 |
| TQA-mc1 | 0.3610 | 0.3770 | 0.3770 | **0.3794** | 0.3684 | 0.3670 | 0.3647 |
| TQA-mc2 | 0.5165 | **0.5412** | 0.5341 | 0.5403 | 0.5387 | 0.5341 | 0.5240 |
| Winogrande | 0.7174 | **0.7285** | **0.7285** | 0.7237 | 0.7238 | 0.7227 | 0.7214 |
| EQ-Bench | 64.901 | 65.372 | **65.583** | 65.246 | 64.974 | 63.268 | 62.373 |

Table 2: Accuracy improvement for each $\lambda$ on all the benchmarks tested at 800 step in PPO. Baseline here is the SFT model, which is LLaMa3-8B-Instruct (Dubey et al., 2024).

## C.2 DETAILED EXPERIMENTAL RESULTS IN SECTION 6.1

In Table 4, we present the comparison results between over-scoring setting and under-scoring setting, where maximum reward and minimum reward are chosen as the reward signal respectively.

The under-scoring (minimum reward) setting consistently outperforms the over-scoring (maximum reward) strategy in most cases. Particularly, in robustness-related tasks such as ANLI, the minimum reward setting yields better stability and accuracy. This suggests that the conservative approach of minimizing rewards helps avoid overfitting to noisy or suboptimal reward signals, which is especially beneficial for tasks that require robust generalization. For benchmarks like GSM8K, which involve complex mathematical reasoning, the minimum reward setting provides a significant advantage, leading to a more stable and gradual improvement in performance. This result aligns with the hypothesis that minimizing rewards helps guide the model through more cautious exploration, preventing drastic policy shifts that could derail learning in tasks requiring precision.

| $\lambda$ | 0 | 0.2 | 0.4 | 0.6 | 0.8 | 1.0 |
|---|---|---|---|---|---|---|
| 200 step | 0.40% | 0.67% | 1.00% | 1.41% | 0.81% | 0.01% |
| 800 step | 2.40% | 2.41% | 2.61% | 2.22% | 1.49% | 0.19% |

Table 3: Mean accuracy improvement for each $\lambda$ on all the benchmarks tested.

Table 4: The performance of PPO with maximum reward vs. PPO with minimum reward.

| Benchmark | Accuracy Range | Category | Min or Max? |
|---|---|---|---|
| ANLI-r1 | 0.45-0.50 | Robustness | Min |
| ANLI-r2 | 0.45-0.50 | Robustness | Min |
| ANLI-r3 | 0.45-0.50 | Robustness | Max |
| ARC-Challenge | 0.50-0.55 | General Knowledge | Min |
| ARC-Easy | 0.80-0.85 | General Knowledge | Min |
| Arithmetic | 0.85-0.90 | Numerical Computation | Equal |
| EQ-Bench | - | Emotion Reading | Min |
| FDA | 0.75-0.80 | Information Extraction | Min |
| GSM8K | 0.30-0.40 | Math Reasoning | Min |
| Lambada | 0.70-0.75 | Context Understanding | Max |
| MMLU | 0.60-0.65 | General Knowledge | Max |
| PIQA | 0.75-0.80 | Commonsense | Min |
| SciQ | 0.95-1.00 | Commonsense | Min |
| TQA-mc1 | 0.35-0.40 | General Knowledge | Min |
| TQA-mc2 | 0.50-0.55 | General Knowledge | Min |
| Winogrande | 0.70-0.75 | Reasoning | Min |

Interestingly, on tasks like ANLI-r3, Lambada, and MMLU, the maximum reward setting leads to better performance. These benchmarks often require a deeper understanding of context or broad general knowledge, where more aggressive optimization via over-scoring might help the model capture more subtle nuances in the data. However, it is important to note that the improvements in these cases are limited, and there is a risk that the model could overfit to specific examples or data patterns in the long run.

Another key observation is that in domains where numerical or commonsense reasoning is critical, such as Arithmetic, PIQA, and SciQ, the performance difference between the maximum and minimum reward settings is either negligible (as seen in Arithmetic) or favors the minimum reward setting. This reinforces the idea that under-scoring generally promotes more consistent and stable learning, particularly in tasks that require logical consistency or factual accuracy.

In tasks like Winogrande and FDA, which involve reasoning and information extraction, the minimum reward setting once again provides superior performance. This highlights the broader applicability of conservative reward strategies, especially in tasks where incorrect reward signals can quickly lead the model astray.

Overall, the comparison shows that while the over-scoring strategy can occasionally provide short-term performance boosts, particularly in context-heavy tasks, the under-scoring (minimum reward) setting generally yields more reliable and stable improvements across a wider range of benchmarks. This underscores the value of conservative reward modeling, especially when dealing with complex reasoning tasks or when the RM itself may be noisy or imperfect. The results further validate the robustness of the reward-robust RLHF approach, where minimizing the reward signal helps mitigate the risk of performance degradation over time.

## C.3 ABLATION STUDY ON BRME

In this section, we present ablation study results on the RMs to highlight the superiority of BRME over the traditional RM. The coverage and margin advantages are detailed in Appendix B.3. Here, we provide a comparison of accuracy and the direct effect on the PPO process.

|  | General QA | Writing | Comprehension | Math | Total |
|---|---|---|---|---|---|
| Bayesian RME | 75.1% | 74.7% | 76.8% | 77.1% | 75.6% |
| Tradition RM | 73.9% | 73.4% | 76.4% | 75.3% | 74.5% |

Table 5: Ablation study: accuracy comparision between BRME and traditional RM.

We first compared the accuracy of the single-head RM trained with MLE loss to the nominal head of the BRME on the preference dataset. The results in Table 5 clearly demonstrate that BRME achieves higher accuracy across most benchmarks, indicating a better ability to distinguish between preferred and non-preferred responses. Specifically, BRME outperforms the traditional RM by a notable margin across different categories such as General QA, Writing, Comprehension, and Math. The overall accuracy of BRME reaches 75.6%, compared to 74.5% for the traditional RM, showing that BRME's ability to model uncertainty provides a tangible advantage in distinguishing between correct and incorrect responses.

In the General QA category, BRME achieves a 1.2% improvement over the traditional RM, and similar improvements are observed in Writing (1.3%) and Math (1.8%). These results demonstrate the robustness of BRME across various types of tasks, where more accurate reward signals are crucial for guiding the optimization process. The largest gap is observed in the Math category, where BRME shows a significant improvement, further reinforcing the model's ability to handle complex reasoning tasks where reward signals from traditional models may be less reliable.

We also examined the impact of these RMs on the PPO process by comparing the performance of actors trained using BRME and the traditional RM. The results indicate that actors trained with BRME benefit from more stable and reliable reward signals, leading to smoother training curves and improved final performance. This is particularly evident in tasks requiring more nuanced decision-making, where BRME's broader reward distribution prevents the actor from overfitting to narrow or incorrect reward signals.

## C.4 ABLATION STUDY ON INTEGRATION STRATEGY

In addition to evaluating the min/max integration strategies, we conducted the same PPO experiment using the mean of all reward signals and compared it to other integration strategies, such as the min, max, and trade-off strategies. The mean strategy represents a baseline approach where all reward signals are averaged, which can smooth out the variability in individual rewards but may also mask valuable distinctions between reward sources. Comparing it with other strategies allows us to assess how different approaches to integrating reward signals impact the stability and performance of the model during training. The results are presented in Figure 9.

The performance of the mean strategy consistently falls between the min and max strategies, indicating that while averaging rewards offers a more balanced approach, it does not capture the advantages of more targeted integration strategies. Specifically, in the later stages of training (after 800 steps), the mean strategy shows a tendency toward performance decline, suggesting that the averaging process may dilute the reward signal over time, leading to suboptimal policy learning. This is especially evident in tasks requiring precise optimization, where overly smoothing the reward signal prevents the model from fully leveraging high-quality responses.

On most datasets where PPO has a significant effect, the reward-robust RLHF setting with a trade-off parameter of $\lambda = 0.6$ outperforms the mean strategy. The trade-off strategy balances nominal performance with robustness, allowing the model to benefit from both high-performing and conservative reward signals. This balance is crucial in long-term training, as it mitigates the risk of overfitting to specific rewards and helps maintain consistent performance gains.

The mean strategy, while easier to implement, lacks the ability to differentiate between high-quality and low-quality reward signals effectively. By averaging all signals, it fails to account for the underlying uncertainty or variance in the RM. In contrast, the reward-robust RLHF approach, particularly with $\lambda = 0.6$, preserves the benefits of more robust exploration and ensures that the model can continue improving in the later stages of training, as observed in the experiments.

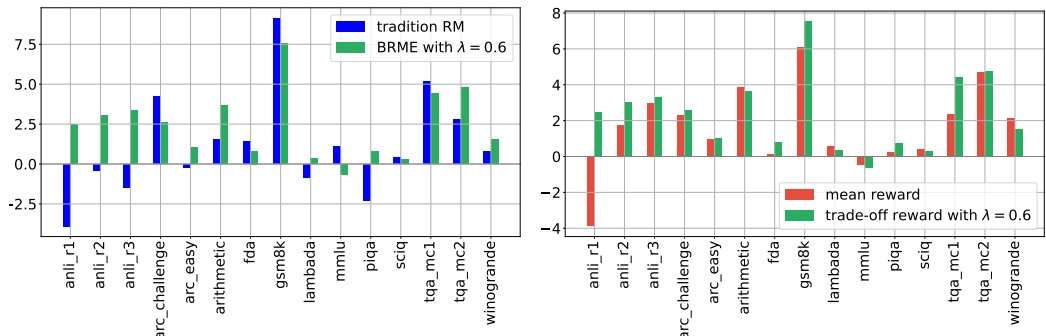

(a) PPO performance comparison between traditional RM and BRME.

(b) PPO performance comparison between Mean integration and trade-off with $\lambda = 0.6$.

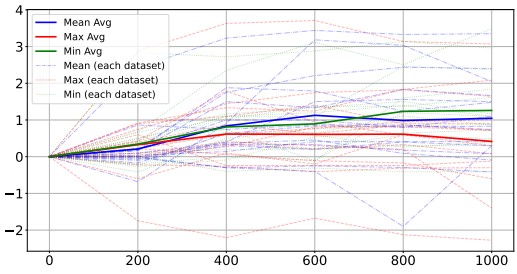

(c) Ablation study compared with mean strategy.

Figure 9: Ablation study on BRME and integration strategy.

Additionally, the decline in performance seen with the mean strategy after 800 steps may be attributed to the lack of adaptability in handling varied and uncertain reward signals. As training progresses, the model requires more refined guidance to navigate complex optimization landscapes, which the mean strategy fails to provide. This explains why the reward-robust approach, which dynamically adjusts the balance between performance and robustness, consistently yields better results in long-term training.

## D    DATASET DESCRIPTIONS

In this section, we provide an overview of the datasets used for training and the benchmarks used for evaluation in our experiments. This includes a detailed description of the datasets utilized in the BRME training process and the PPO training pipeline, as well as the benchmarks employed to evaluate the performance of our models.

Table 6 outlines the various datasets used for training BRME and the PPO models. These datasets span a wide range of task types, ensuring that the RM is exposed to diverse examples during training, helping it generalize across different domains. In UltraFeedBack, we selected response pairs where the score is greater than or equal to 2 to form the chosen and rejected response pairs, which were then used to train the BRME. By focusing on high-quality response pairs, we aim to ensure that the RM is robust and capable of accurately distinguishing between better and worse responses.

For evaluation, we utilized a comprehensive set of benchmarks, as detailed in Table 7. These benchmarks cover various domains, including general knowledge, logical reasoning, commonsense, and mathematical reasoning, allowing us to thoroughly assess the performance of the RMs across different types of tasks. To ensure consistency and reliability in our evaluation process, we employed the LM-evaluation-harness framework, which is a widely-used standard for evaluating LLMs (Gao et al., 2024). This framework provides a standardized and rigorous approach to comparing model performance across a diverse set of tasks, ensuring that the results are comparable with other works in the field.

Table 6: Description of the datasets in the training pipeline.

| Datasets | Size | Description |
|---|---|---|
| HH-RLHF | ~170,000 | The HH-RLHF dataset is a collection designed to train and evaluate language models on human preferences, particularly focusing on making AI models both helpful and harmless. |
| UltraFeedBack | 336,820 | The UltraFeedback dataset is a large-scale, high-quality, and diversified preference dataset designed to enhance the performance of RLHF. It includes over 1 million feedback instances generated by GPT-4 for around 250,000 user-assistant conversations across various aspects of language model outputs, such as helpfulness, truthfulness, and honesty. |
| Internal Dataset | 128,508 | The internal dataset is collected and filtered by the PM team. The categories includes general knowledge, numerical computation, reasoning, person writing, etc. |

By using a diverse set of training datasets and evaluation benchmarks, we aim to provide a comprehensive assessment of our model's capabilities. The combination of varied training data ensures the robustness of the RM, while the evaluation benchmarks test the model's ability to generalize across different domains, making our findings relevant to a wide range of applications.

## E  PROOF FOR LEMMA 1

In this section we first reclaim the Lemma 1, and provide the proof details.

**Revisiting Lemma 1.**  Let the RM provide a constant reward $r(s, a) = c$ for all actions $a \in \mathcal{A}$ and states $s \in \mathcal{S}$ during PPO training. Then, the gradient of the PPO objective function with respect to the policy parameters $\theta$ is zero, implying that the actor cannot be optimized under such a RM.

*Proof.* During PPO training, the objective function is given by:

$$\ell^{\text{PPO}}(\theta) = \mathbb{E}_t \left[ \min \left( r_t(\theta) \widehat{A}_t, \text{clip}(r_t(\theta), 1 - \epsilon, 1 + \epsilon) \widehat{A}_t \right) \right],$$

where $r_t(\theta) = \frac{\pi_\theta(a_t|s_t)}{\pi_{\theta_{\text{old}}}(a_t|s_t)}$ is the probability ratio between the current policy and the old policy, and $\widehat{A}_t$ is the advantage function. The advantage function is defined as:

$$\widehat{A}_t = Q(s_t, a_t) - V(s_t),$$

where $Q(s_t, a_t)$ is the state-action value function and $V(s_t)$ is the state value function, both output by the critic model.

If the RM provides a constant reward $r(s_t, a_t) = c$ for all actions, the state-action value function and the state value function become:

$$Q(s_t, a_t) = \frac{c}{1 - \gamma}, \quad V(s_t) = \frac{c}{1 - \gamma},$$

where $\gamma$ is the discount factor. Therefore, the advantage function simplifies to:

$$\widehat{A}_t = Q(s_t, a_t) - V(s_t) = \frac{c}{1 - \gamma} - \frac{c}{1 - \gamma} = 0.$$

Substituting this into the PPO objective function, we get:

$$\ell^{\text{PPO}}(\theta) = \mathbb{E}_t \left[ \min \left( r_t(\theta) \cdot 0, \text{clip}(r_t(\theta), 1 - \epsilon, 1 + \epsilon) \cdot 0 \right) \right] = 0.$$

Table 7: Description of the benchmarks.

| Benchmark | Size | Description |
|-----------|------|-------------|
| ARC | 7,787 | Grade-school science exam questions, with 2,590 hard ones and 5,197 easy ones. |
| LAMBADA | 5,153 | A dataset to evaluate the capabilities of text understanding by means of a word prediction task. A collection of narrative passages sharing the characteristic that human subjects are able to guess their last word if they are exposed to the whole passage, but not if they only see the last sentence preceding the target word. |
| PIQA | 3,124 | Question Answering (PIQA) is a physical commonsense reasoning to investigate the physical capability of existing models. |
| SciQ | 13,679 | The SciQ dataset contains 13,679 crowdsourced science exam questions about Physics, Chemistry and Biology, among others. An additional paragraph with supporting evidence for the correct answer is provided. |
| WinoGrande | 44,000 | A fill-in-a-blank task with binary options, the goal is to choose the right option for a given sentence which requires reasoning. |
| TruthfulQA | 817 | A QA task aimed at evaluating the truthfulness and factual accuracy of model responses. |
| MMLU | 28,128 | Knowledge-based multi-subject multiple choice questions for academic evaluation. |
| GSM8K | 1,000 | A benchmark of grade school math problems aimed at evaluating reasoning capabilities. |
| FDA | 551 | Tasks for extracting key-value pairs from FDA documents to test information extraction. |
| EQ-Bench | 171 | EQ-Bench is a benchmark for language models designed to assess emotional intelligence. It uses a specific question format, in which the subject has to read a dialogue then rate the intensity of possible emotional responses of one of the characters. Every question is interpretative and assesses the ability to predict the magnitude of the 4 presented emotions. |
| Arithmetic | 2,000 | A small battery of 10 tests that involve asking language models a simple arithmetic problem in natural language. |
| ANLI | 3,200 | Adversarial natural language inference tasks designed to test model robustness. It collected via an iterative, adversarial human-and-model-in-the-loop procedure. It consists of three rounds that progressively increase in difficulty and complexity. |

As a result, the gradient of the objective function with respect to the policy parameters becomes:

$$\frac{\partial \ell^{\text{PPO}}(\theta)}{\partial \theta} = 0,$$

which implies that no update to the actor's parameters will occur, and the actor will not be optimized. □

## F  FUTURE WORKS: HETEROLOGOUS REWARD FUSION

Heterologous Reward Fusion (HRF) aims to enhance the robustness and coverage of the uncertainty set in our RMs by incorporating multiple heterologous reward sources. This method involves combining several different RMs trained on diverse datasets, such as Baichuan2-33B (Yang et al., 2023), Qwen2-72B (Yang et al., 2024a), and LLaMa3-8B (Dubey et al., 2024). The key advantage of using

Table 8: Reward distribution characteristics in several RMs trained from different base model.

| Base Model | Size | Max | Min | Mean | Std | Acc |
|---|---|---|---|---|---|---|
| LLaMa3 | 8B | 15.625 | -14.687 | -0.032 | 4.529 | 0.789 |
| Baichuan2 | 13B | 10.562 | -13.062 | 2.024 | 2.699 | 0.868 |
| Baichuan2 | 33B | 7.125 | -7.093 | 0.573 | 2.595 | 0.917 |
| Qwen2 | 72B | 16.625 | -10.062 | -0.978 | 3.126 | 0.934 |
| Baichuan2 | 177B | 14.187 | -6.687 | 6.697 | 2.836 | 0.950 |

heterologous models lies in the diversity of their training data, which produces more varied reward scores and helps to enrich the uncertainty set.

By integrating heterologous rewards, we aim to capture a broader spectrum of reward signals, leveraging the strengths of models trained on different datasets and with varied optimization goals. For example, integrating direct scoring from closed-source LLM APIs like GPT-4 alongside open-source models ensures a wider and more balanced reward distribution, implicitly utilizing data from diverse sources. This heterogeneity allows for a more comprehensive assessment of the model's performance across different scenarios.

However, one of the primary challenges in HRF is that each reward source has different value ranges, making direct comparisons potentially unfair. To address this, we perform empirical reward normalization. We score a separate dataset, HH-RLHF (Bai et al., 2022a), using each reward source and compute the mean and variance for each. During PPO training, we transform the rewards from each source by normalizing them based on their respective means and stds. This normalization helps ensure that the reward signals from different models are comparable, allowing for a more fair integration of the rewards.

Table 8 presents the results of our HRF experiment, showing the reward distribution characteristics (max, min, mean, and std) for each model, along with their accuracy. Accuracy here is defined as the proportion of instances where the model correctly identifies the chosen response as superior to the rejected one.

However, it is important to note that we did not conduct end-to-end PPO experiments incorporating the full RM pipeline in this exploration. The primary reason for this limitation is the significant computational resources and time required to carry out such experiments comprehensively. Running full-scale PPO experiments, especially when integrating multiple heterologous RMs, is computationally expensive and requires extended periods of training, particularly when dealing with large models like Baichuan2-33B and Qwen2-72B. Moving forward, our next steps include performing the full end-to-end PPO experiments to evaluate the impact of HRF on the performance of the trained policy.

## G  LIMITATIONS

While the proposed reward-robust RLHF framework shows improved performance on automatic evaluation benchmarks, several non-negligible limitations remain. First, for some prompts, if all reward heads in the BRME fail to provide correct signals and exhibit similar error patterns, the reward hacking behavior seen in standard RLHF cannot be entirely mitigated, even with the reward-robust approach. This suggests that the ultimate optimization still heavily depends on the model's capacity and generalization ability, particularly in handling out-of-distribution (OOD) data. As a result, improving the quality of the reward model and the data used to train PPO is likely to be the most critical factor influencing long-term training success.

Second, some of the theoretical hypotheses proposed to explain the experimental results in Section 6.1 still require further validation. Conducting fine-grained analysis, particularly regarding the specific impact of over-scoring and under-scoring signals on the PPO exploration process, will be highly valuable. Addressing these gaps will be one of our future research focuses.

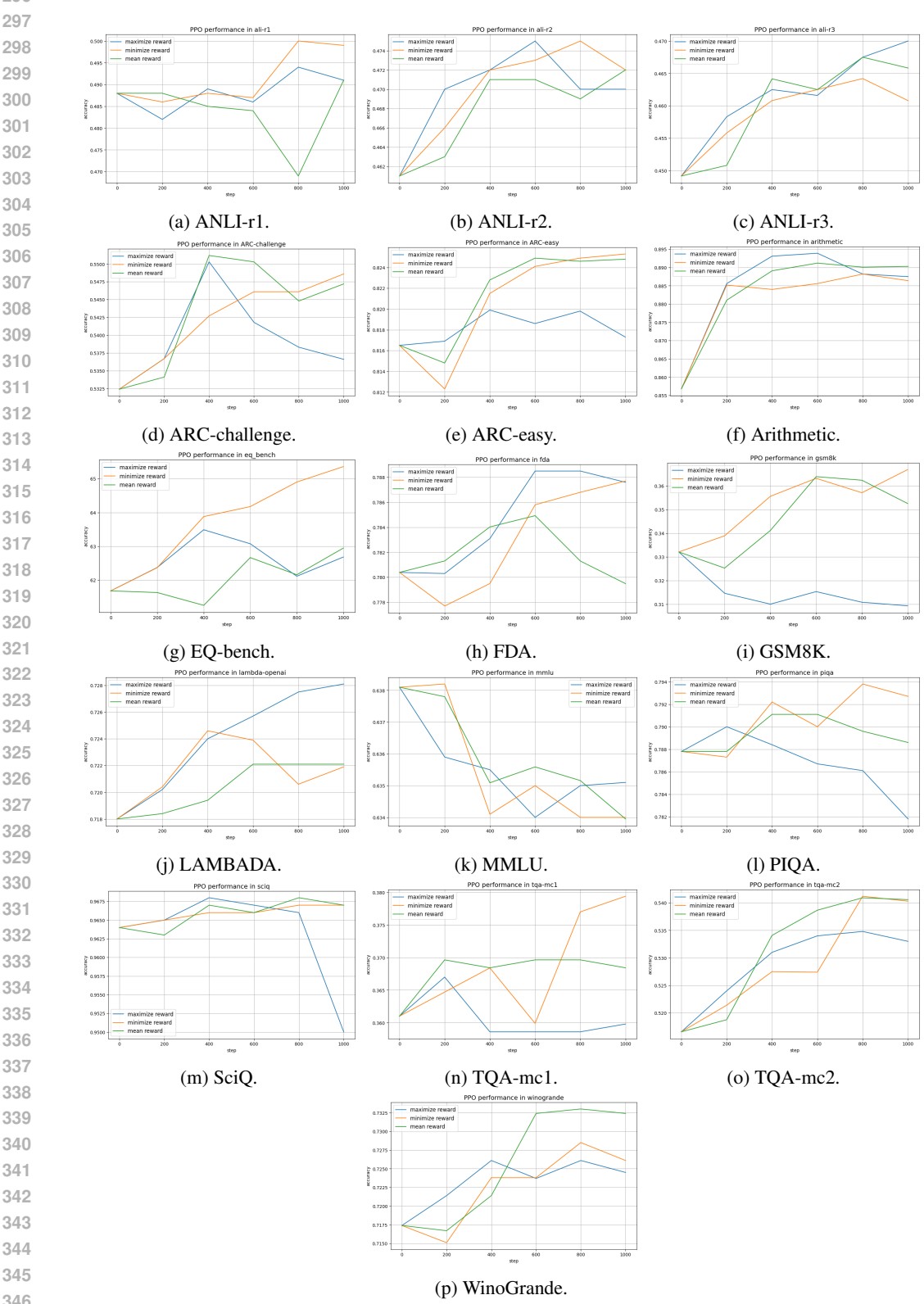

(a) ANLI-r1.  (b) ANLI-r2.  (c) ANLI-r3.

(d) ARC-challenge.  (e) ARC-easy.  (f) Arithmetic.

(g) EQ-bench.  (h) FDA.  (i) GSM8K.

(j) LAMBADA.  (k) MMLU.  (l) PIQA.

(m) SciQ.  (n) TQA-mc1.  (o) TQA-mc2.

(p) WinoGrande.

Figure 10: Performance comparison between the PPO performance in maximized reward, minimized reward and the mean reward.

