# OpenReview forum: "Reward-Robust RLHF in LLMs"
_ICLR.cc/2025/Conference — Submitted to ICLR 2025_

### Official Review · Reviewer_8ZLZ · 2024-10-20

**Soundness:** 4
**Presentation:** 3
**Contribution:** 3
**Rating:** 6
**Confidence:** 3

**Summary:**

This paper proposes a reward modeling method that aims to be robust despite what the paper refers to as inherent imperfections of reward models. To motivate their method, a toy model is constructed in which they do reward modeling with perfect preference data (i.e., there have been no labeling mistakes). They show that even with ideal data reward modeling is imperfect, which should be expected to worsen in a realistic setting.

Their method has two main components. The first is a technique for combining a conservative reward with a nominal reward. For the conservative reward, the lowest reward from a set of reward models is used. The overall objective is a weighted sum of the conservative and nominal rewards, controlled by a parameter λ. The second component is a reward model architecture where a shared 'base model' is first trained with a standard MLE loss. After a linear layer, there are multiple heads that each output a mean and standard deviation, forming a Gaussian distribution for each head. The head with the smallest standard deviation is the nominal reward, and the head with the lowest mean is the conservative reward.

The paper shows that combining these rewards can improve downstream policy model performance improvements over just using the nominal or conservative reward. They also show that the nominal reward of their reward model architecture causes better downstream policy model performance than a single head reward model, even without considering the conservative reward.

In another experiment, the paper shows that over-scoring the reward seems to be more harmful than under scoring, which supports their inclusion of the conservative reward.

**Strengths:**

- To my knowledge BRME is an original contribution, and the paper motivates the design and use of BRMEs well (e.g., via the toy model experiment).
- BRMEs seem convenient enough that a practitioner would consider using one, and when coupled with the combination of the nominal and conservative reward, the improved downstream performance of the policy model could justify this.
- In general the paper is clear. There are some small grammatical mistakes, but they did not hinder my understanding of the paper. The figures are also descriptive and helpful.
- The appropriate ablation experiments were performed. I do not have concerns about soundness.
- The experimental details in the Appendix are thorough and I do not have concerns with reproducibility, although an open source implementation of BRME would be valuable.

**Weaknesses:**

- The performance gains from using BRME over a single head reward model seem modest (1.1% averaged over the benchmarks) despite the architecture being significantly more complex than a standard reward model.
- Combining rewards has been done in the robust RL literature before (including where one of the rewards is more conservative), and to my understanding contributes more to the performance improvement than the specifics of BRMEs. However, this is applied in a way that seems particularly well suited to BRMEs.

**Questions:**

Could you give a comparison of BRME with a simpler reward model ensemble, or otherwise show how much of the performance gain can be attributed to the combination of nominal and conservative rewards versus the use of this combination with a BRME?

---

> ### Author Response · Authors · 2024-11-24
>
> Thank you for the constructive comments. Below we address the detailed comments.
>
> **Q1: Could you give a comparison of BRME with a simpler reward model ensemble, or otherwise show how much of the performance gain can be attributed to the combination of nominal and conservative rewards versus the use of this combination with a BRME?**
>
> **R1:** Thank you for bringing it up. ***In Appendix D.4, we have done the ablation study on integration strategy, including mean integration, in another word, a  simpler reward model ensemble.*** The result can be found in Figure 9.b and Figure 9.c.
>
> In Appendix D.4, line 1076-1112, we have made a thorough comparison. In general, the mean strategy does not effectively distinguish between high-quality and low-quality reward signals, ignoring uncertainty in the RM. This leads to performance decline after 800 steps due to its inability to adapt to varied reward signals. In contrast, the reward-robust approach, especially with $\lambda = 0.6$, preserves robust exploration and provides more refined guidance, resulting in consistently better performance in long-term training.

---

> > ### Comment · Reviewer_8ZLZ · 2024-11-25
> >
> > Thank you for the reply. As I understand it, Appendix C.4 makes two comparisons:
> > 1. A traditional reward VS BRME. This is different from my request as I was hoping for a comparison between the $\lambda$ = 6 reward model and BRME.
> > 2. $\lambda$ = 6 and mean reward. This is different from my request as it does not include BRME.
> >
> > I have made an effort to compare BRME from the first figure in Appendix C.4 with the $\lambda$ = 6 results in the second figure. If you could compare these and show a significant improvement over $\lambda$ = 6 with BRME, I will consider revising my score upward.

---

> ### Author Response · Authors · 2024-11-25
>
> **2. $\lambda=0.6$ and mean reward. This is different from my request as it does not include BRME.**
>
> Thank you for the feedback. Could the reviewer please clarify why you believe this does not include BRME? Both of these experiments were conducted on BRME, with the difference being the integration strategy. With the robustness integration, the standard deviation is used to determine the nominal reward, incorporating a robustness term in a trade-off manner. In contrast, the mean integration simply outputs the mean of all score heads, without utilizing the uncertainty measurement.
>
> We are more than willing to provide additional experimental results to address the reviewer's concerns, and we would greatly appreciate any further guidance on what specific experiments you have in mind. Currently, we assume the suggestion is to train a new RM ensemble using MLE and apply mean integration as a baseline. However, our approach fundamentally relies on leveraging uncertainty, which cannot be modeled with an MLE-based RM ensemble. Moreover, as mentioned in our response to Reviewer joB9, the two-stage training process can be interpreted as model distillation on the reward model, making the pure mean of BRME not significantly different from a new RM ensemble trained solely using MLE.

---

### Official Review · Reviewer_joB9 · 2024-11-01

**Soundness:** 2
**Presentation:** 2
**Contribution:** 3
**Rating:** 5
**Confidence:** 4

**Summary:**

The authors propose modifying the RLHF objective by introducing an ensemble of reward functions. When running PPO, the minimum of the predicted reward values is used.

**Strengths:**

* The general idea makes sense: do robust learning on the reward function, because there is often a lot of uncertainty over rewards.
* The authors test their method on a wide variety of benchmarks.
* As far as I know, applying robust optimization over the reward function for RLHF is a new combination not used before.

**Weaknesses:**

A good evaluation of "robust" optimization for RLHF would be useful to the community. However, at present the draft is a bit confusing, and I have a number of questions that I think need better explanation and in-text edits for the paper to cross the bar of publication.
Specifically, there are several design choices which are not fully explained or ablated, and the method seems a bit complex. You have to train one reward model the standard way, then another ensemble with MSE. The authors haven't shown evidence that this step is important.
Then, you need to additionally tune the lambda hyperparameter.

Weaknesses with respect to the method:
* The approach seems to be rather involved -- requiring the training of multiple different RMs. First a Bradley-Terry RM, then one trained with MSE.
* Why do the authors have to do this with the MSE loss? It is unclear to me why an ensemble of reward functions trained with the bradley terry model ("MLE" in the authors words) would not demonstrate the same trend.
* I don't udnerstand L791 and eq 8. How is the reward equal to the "mean" plus the action times a variance. The action is a vector of tokens and the reward is a scalar. Where is the first reward model used? does it output the mean? How is the standard deviation chosen?
* Why is this method "bayesian"? Changing the output to a distribution is still consistent with a frequentist perspective -- the authors say they use MLE on line 297. What prior are the authors using? Can the authors make the connection to being "bayesian" more clear as it is the name of their method?
* Why is the uncertainty set of size n? When did the authors define the uncertainty set? Are there results for different sizes of n? Is there a point where making n too large makes you too conservative? It seems like an ablation is missing here.
* The math presented in the paper does not cross the bar for being considered "theory" in my mind. I would encourage the authors to restructure this.

Weaknesses with respect to the experiments:
* I dont understand why evaluating on benchmarks like MMLU predictive of reward overoptimization or uncertainty, the problem the authors claim to be addressing, especially given the feedback data the authors use does not appear to be for these benchmarks.
* While I appreciate the authors time spent running the stochastic case analysis, I don't think I fully understand its purpose right now. Why would training LLMs using random reward functions provide insights about different methods? What would we want to see?
* 1000 Steps of PPO seems really small. Is this a standard choice? Sometimes when finetunig with RL performance goes down before going back up again.
* Fig 3: What is difference in scores calculated with respect to? Is it just the base model before RLHF?


## Edits
* L37 "on a preference data"
* I'm not sure it has been proven that RLHF is a key pathway to "achieveing AGI". I would personally calm down this language.
* Is it known that OpenAI o1 used RLHF? I was under the impression it was more  "RL" than "HF"?
* Line 49: This is confusing. The manuscript went from talking aobut optimization challenges to what seems like a bias problem in data. How does robust RLHF improve that?
* L94: Citations -- training a reward model on preferences was done wayyy back by Arkour (2004), Sadigh (2017), and more. This statemetn should be about language modeling or include more citations.
* L96: "widely recognized that these methods are suboptimal" is a bit strong.
* L102: How does PEBBLe by Lee et al show that diversity constraints lead to overfitting / underfitting problems? From what I understand, Lee et al. show that unsupervised pretraining is important for *online* RLHF so the initial sampled comparisons aren't of useless behaviors.
* L128: Does distributional RL not also belong here as a form of RL that considers the whole distribution of rewards?
* L147: How does Kwon et al. show that "uncertainty in the reward model" impacts final performance? From what I understand, Kwon et al use a pretrained LLM as the reward model for a game-playing agent.
* L182: Though as written it is still correct, its a bit strange that loss functions are written without the average, making their magnitudes dataset-size dependent.
* L245: "neglect of the uncertainty set" makes it sound like the uncertainty set is always there, but it seems like it is something the authors expliclty learn.
* L454: I don't think this deserves to be a Lemma.

**Questions:**

* Why is the authors method bayesian? Can this connection to the title be expalined the word bayes is barely mentioned in the text.
* Why are evaluations on MMLU a good proxy for reward overoptimization from human feedback data? Do I mis-understand the experiemnts? Clarification would be appreciated.
* I don't understand L791 and eq 8. How is the reward equal to the "mean" plus the action times a variance. The action is a vector of tokens and the reward is a scalar. Where is the first reward model used? does it output the mean? How is the standard deviation chosen?
* Why are all the phases of reward training necessary?
* several more questions were posed in the weaknesses section.

---

> ### Comment · Reviewer_joB9 · 2024-11-22
> **Seeking additional clarification**
>
> Could the authors comment on the difference between their work and this paper: https://arxiv.org/abs/2310.02743, published at ICLR last year that uses what appears to be a similar objective, up to the mixing coefficient $\lambda$.

---

> > ### Author Response · Authors · 2024-11-24
> >
> > **Q12: The difference between our work with** https://arxiv.org/abs/2310.02743
> >
> >
> >
> > **R12:** Thank you for the question. We conclude the difference as follows briefly,
> >
> > - **Different theoretical basis**: Our work is grounded in the robustness-performance trade-off studied in prior research on RL/MDP, whereas the main focus of the referenced work is on the ensemble efficiency of reward models.
> >
> > - **Different method proposed**: The core of our approach is the modeling, calculation, and utilization of the uncertainty set, which is not addressed in the other work. We select the nominal reward based on uncertainty measurement and employ a trade-off ensemble, while the other work primarily focuses on mean, max, and min ensemble methods. Additionally, the training pipeline of our reward model differs from that of the other work.
> >
> > - **Different experimental results:** Our method consistently achieves better performance compared to mean/min/max ensembles across various benchmarks, whereas in the other work, the performance of the ensemble methods is more variable.
> >
> > - **Additional contribution**: We provide theoretical insights into reward model imperfections, demonstrating through a synthetic toy model that even with an ideal annotator, achieving a perfect reward function is impossible. We also show that under-scoring is preferable to over-scoring in long-term training, given the inherent bias in accessible rewards. Furthermore, we conduct a stochastic case analysis, demonstrating that our proposed framework remains effective in out-of-distribution scenarios, with the robustness regularization term stabilizing the training process by narrowing the reward distribution.

---

> ### Author Response · Authors · 2024-11-24
>
> Thank you for the constructive comments. Below we address the detailed comments.
>
> **Q1: Why do the authors have to do this with the MSE loss and why would not MLE demonstrate the same trend? Explain the necessity of the two-stage RM. The approach seems to be rather involved**
>
> **R1:** Thank you for the question. Let us clarify why the first stage, which trains a standard RM, is necessary. The standard RM trained with BT model and MLE serves as a numerical benchmark, calculating the probability that $a^+$ is better than $a^-$ for each preference pair ($\hat{p}(a^+ \succ a^-)$). However, since this produces only a scalar value, it cannot indicate the model's confidence in its judgment. Therefore, we introduce a second stage using MSE loss to train the ultimate BRME, as shown in equation (8) in Appendix B.1. We demonstrate in Appendix B.1 that the output standard deviation reflects model uncertainty: as more optimization steps occur in a specific data domain, the model becomes more confident in its evaluation, resulting in a smaller standard deviation.
>
> For better understanding, one can think of the two-stage training process as model distillation, where the final RM inherits the standard RM's capabilities and gains the ability to measure uncertainty. Several works have used similar training techniques [1,2], which we recommend to the reviewer.
>
> In terms of complexity, we only need to make a few changes to the standard training pipeline, and the method can be extended to various types of RMs, algorithms and training frameworks Without violating the anonymity clause, we will open source the RM training process in the near future.
>
> *[1] Wu Y, Sun Z, Yuan H, et al. Self-play preference optimization for language model alignment[J]. arXiv preprint arXiv:2405.00675, 2024.*
>
> *[2] Lou X, Yan D, Shen W, et al. Uncertainty-aware reward model: Teaching reward models to know what is unknown[J]. arXiv preprint arXiv:2410.00847, 2024.*
>
> **Q2: The explanationHow is the reward equal to the "mean" plus the action times a variance? (L791 and eq 8)**
>
> **R2:** Thank you for your question. The reparameterization trick is a commonly used technique to handle non-differentiability in sampling processes (especially in methods like VAE and Bayesian NN), which allows us to differentiate through stochastic variables. Specifically, without reparameterization, directly sampling from a stochastic distribution (in this case, the reward distribution) introduces non-differentiability, making it challenging to compute gradients and perform backpropagation.
>
> In our case, we employ the reparameterization to express the sampled reward as a deterministic transformation of a stochastic variable. By representing the reward as Equ.8, we essentially rewrite the stochastic sampling operation in a form that allows gradients to be propagated through $\mu_i^+$, $\mu_i^-$, $\sigma_i^+$, and $\sigma_i^-$. This approach transforms the sampling into a combination of differentiable operations (i.e., adding the mean and scaling the variance by a stochastic component), allowing us to update these parameters effectively during optimization.
>
> **Q3: Where is the first reward model used? does it output the mean? How is the standard deviation chosen?**
>
> **R3:** The first reward model generates the probability that $a^+$ is better than $a^-$ for each preference pair ($\hat{p}(a^+ \succ a^-)$). The standard deviation is the model output, from each head, and is the value we optimize on.
>
> **Q4: Why is this method "bayesian"?**
>
> **R4:**  Thank you for your question. The reward generated by the first-stage reward model, represented by $\hat{p}(a^+ \succ a^-)$, serves as an implicit prior. In the second stage, we refine this prior by using additional training data, aligning with the Bayesian perspective of updating an initial belief based on new evidence.
>
> Specifically, the first stage provides point estimates through MLE, which we consider as prior beliefs. In the second stage, we model the reward as a Gaussian distribution with mean $\mu_i$ and standard deviation $\sigma_i$, incorporating uncertainty around these initial estimates. This distributional approach transforms the prior point estimates into a posterior distribution, representing the updated belief about the rewards.
>
> We have clarified this Bayesian perspective in the revised manuscript to ensure the connection is more explicit in Appendix B.1.

---

> ### Author Response · Authors · 2024-11-24
>
> **Q5: Why is the uncertainty set of size n? When did the authors define the uncertainty set? Are there results for different sizes of n? Is there a point where making n too large makes you too conservative?**
>
> **R5:** Thank the reviewer for bringing it up. The size of the uncertainty set is set to be 5 in our paper and it is an empirical best choice due to our prior experiments, and is aligned with [1,2] where 5 models are ensembled.  The definition of the uncertainty set is in Section 5, line 239-242.
>
> To our best knowledge, it is unusual to use an extreme large number of ensembles in model ensembling. But with a larger uncertainty set, the policy is highly possible to be too conservative, but it didn’t happen in our setting due to the trade-off. We will leave the exploration of the scaling effect of the uncertainty set as a potential future work.
>
> *[1] Coste T, Anwar U, Kirk R, et al. Reward model ensembles help mitigate overoptimization[J]. arXiv preprint arXiv:2310.02743, 2023.*
>
> *[2] Lou X, Yan D, Shen W, et al. Uncertainty-aware reward model: Teaching reward models to know what is unknown[J]. arXiv preprint arXiv:2410.00847, 2024.*
>
> **Q6: The math presented in the paper does not cross the bar for being considered "theory"**
>
> **R6:** Thank you for the suggestion. We have rephrased it in the revision.
>
> **Q7: Why are evaluations on MMLU a good proxy for reward overoptimization from human feedback data?**
>
> **R7:** MMLU (as well as other benchmarks) just serves as a coomon benchmark to evaluate the model’s foundamental capabilities, and has been incorporated in many related works in RLHF [3,4]. We choose it to an end-to-end evaluation of the proposed method, rather than a specific proxy for evaluating reward overoptimization.
>
> *[3] Gao B, Song F, Miao Y, et al. Towards a unified view of preference learning for large language models: A survey[J]. arXiv preprint arXiv:2409.02795, 2024.*
>
> *[4] Jiang R, Chen K, Bai X, et al. A Survey on Human Preference Learning for Large Language Models[J]. arXiv preprint arXiv:2406.11191, 2024.*
>
> **Q8: Why would training LLMs using random reward functions provide insights about different methods?**
>
> **R8:** The random reward setting simulates a scenario where the RM encounters a data domain that it wasn’t optimized on and fail to give right reward direction, which is very common in practice. It can be considered as a worst-case analysis where the robustness trade-off plays its role. By accounting for uncertainty in the RM, the framework effectively narrows the optimization space, allowing the model to avoid over-optimization based on potentially erroneous reward signals.
>
> **Q9:  1000 Steps of PPO seems really small. Is this a standard choice? Sometimes when finetunig with RL performance goes down before going back up again.**
>
> **R9:** Yes, it is quiet standard to optimize 600-800 steps in the LLM alignment in our practice. 1000 steps can be considered as a long-run training given the challenge of keeping RLHF in a stable pace. The batch-size is 128, so over 128000 data points were consumed. The going down and up process happens within these 1000 steps in many benchmarks, and 800 steps nearly reaches the covergence phase. We refer the reviewer to the experimental results in Figure 9.
>
> **Q10: What is difference in scores calculated with respect to? Is it just the base model before RLHF?**
>
> **R10:** It is the performance improvement in RLHF, given the SFT model before RLHF as a reference.
>
> **Q11: Some suggestions on the writing.**
>
> **R11:** We have edited the manuscripts according to the advice (blue highlight in the revision). Here are some specific points to clarify:
>
> - Is it known that OpenAI o1 used RLHF? I was under the impression it was more "RL" than "HF"?
>
> The subject of the sentence is RL, not RLHF.
>
> - Does distributional RL not also belong here as a form of RL that considers the whole distribution of rewards?
>
> While distributional RL indeed considers the entire distribution of rewards, its primary focus is on modeling the variability in reward outcomes rather than directly tackling robustness against uncertainties like transition, observation, action, or disturbance.  So we didn’t list it.
>
> - L454: I don't think this deserves to be a Lemma.
>
> The lemma is straightforward and describe the ideal situation, but the intention is to draw attention to Remark 1, which highlights the non-ideal dynamics of the model optimization process and how the KL term influences convergence behavior.

---

> ### Comment · Reviewer_joB9 · 2024-11-30
> **Thanks for the response**
>
> I'd like to thank the authors for their response!
>
> **Q1: Using the "MLE Model"**
> The authors explanation generally makes sense -- thanks!
>
> **Q2: Reparameterization**
> Yes, I realize this is the reparameterization technique, but I think the notation used in the paper is rather confusing. I woulid suggest improving it. For example, the mean in this case is produced by the model (so maybe it should be conditioned on the text?)
>
> **Q3**
> Thanks for the clarification
>
> **Q4**  The details about using a different dataset for training the BRME seem important -- this should be a bigger point in the main paper! I still find this justification of being bayesian-ness a bit weak, particularly as it depends on this set of additional training data which (to my knowledge after another quick scan of the paper) doesn't seem to be in the main text. I really think that the details / justification for why the method is bayesian should be more central to the paper, as "bayesian" is in the title!
>
> **Q5**. This would have been a nice ablation, but I understand the authors may not have time/ resources for it.
>
> **Q7** I understand that MMLU is a common benchmark for question/answering capabilities, but if the preference datasets are not focused on QA capabilities, its unclear why we can measure MMLU to determine how well preference learning worked. IE consider a scenario where doing really well on your preference dataset hurts MMLU. Then, methods which actually do the worst at RLHF might do better. Is there precedent for using MMLU to measure how good RLHF performance is with UltraFeedback? I understand that MMLU is a standard benchmark (as stated by the references the authors linked), but my question is more subtle: why does MMLU (distribution X) measure how well we did at optimizing Ultrafeedback (distribution Y)? For example, https://arxiv.org/abs/2310.02743 trains and validations on the Alpaca dataset, which means they are measuring the same thing.
>
> **Q9** Thanks for the response. I'm more familiar with PPO in the continuous control setting where > 100k steps are standard.
>
> **Q10** Thanks! Please clarify this in the draft.
>
>
> I would also like to thank the authors for discussing the differences between their work and https://arxiv.org/abs/2310.02743. Given these works are very similar, in the final version o the paper it would be nice to have a discussion on the difference, and best to have a comparison, since they also use ensembles to curb over-optimization.
>
> I have raised my score, but concerns remain on:
> * The presentation of the method
> * The experimental evaluation
> * comparisons given the closeness to https://arxiv.org/abs/2310.02743. I know it is too late for the authors to run experiments, so perhaps the inclusion of a discussion of the difference would be good.

---

### Official Review · Reviewer_JCaz · 2024-11-03

**Soundness:** 1
**Presentation:** 1
**Contribution:** 1
**Rating:** 1
**Confidence:** 4

**Summary:**

The paper proposes a reward robust RL objective for RLHF alignment of lLMs

**Strengths:**

1. The paper addresses an important problem of being robust of uncertainty of learned reward functions.

**Weaknesses:**

1. Soundness and Writing: The paper lacks a principled approach or theoretical soundness. The paper is full of unsubstantiated claims, a few examples   in chronological order being "line 10: RLHF is a key pathway towards AGI", "Line 24: Reward Robust RLHF approaches stability of constant reward setting", "Line 36: RLHF leading to breakthrough of o1 models, line 48: challenge of both overfitting and underfitting in reward models", "Line 202: Obtaining an ideal RM is almost impossible".
Every statement made in the paper should be substantiated with evidence through a citation, theory or empirical evidence. The paper currently flows as a article rather than a research paper.

2. Improvements on prior approaches: Authors claim in line 147 that prior methods rely on empirical uncertainty estimation approaches. The methods authors propose is still empirical and without any guarantees.

3. Potentially wrong experimental protocol: The toy example in the paper, does not seem to explain that obtaining an ideal reward model is impossible. The situation in the experiment is artificially created to make sure that reward model is hard to learn. More details about experimental protocols are needed in the main paper.
For the main experiment, the experimental protocol is very sparsely explained. For example, what are 800 steps of PPO? are they data points or gradient steps? Where weren't standard training runs for fixed number of epochs done. Why are we comparing performances between epochs before the algorithm has seen all the training data?

4. Proofs that do not explain the phenomena: The proof in section 6.2 dubbed "Stochastic case analysis" are just trivial proofs that RL will not learn with constant reward. It is not clear how they are related to the method proposed in the paper.

**Questions:**

See weaknesses above.

---

> ### Author Response · Authors · 2024-11-21
>
> Thank you for the comments. Below we response to the detailed comments.
>
> **Q1: Concerns about the potential unsubstantiated claims.**
>
> **R1:** We acknowledge the reviewer's concern about the wordings and has revised some of them with more cautious expression. In line 10, we change the original sentence to "As Large Language Models continue to advance, Reinforcement Learning from Human Feedback is increasingly regarded as a promising approach for enhancing their capabilities and achieving more sophisticated forms of intelligence", avoiding mentioning AGI. In line 36, we We also added citations about the challenges of overfitting and underfitting in RM.
>
> However, we would like to clarify for the rest statements and avoid potential misunderstanding. In our paper, there are clearly corresponding evidence and supports:
>
> - *Line 24: Reward Robust RLHF approaches stability of constant reward setting.*  In Section 6.2 (and the corresponding part in Appendix E), we have shown that in the constant reward setting, the performance of RLHF is acceptable and the proposed method mitigates the instability even with the pure random reward signal.
>
> - *Line 36: RLHF leading to breakthrough of o1 models.* We didn't mention RLHF leads the breakthrough of o1. The subject of the sentence is RL and it is a well-known fact according to existing public reports.
>
> - *Line 202: Obtaining an ideal RM is almost impossible.* We believe that this is a self-evident reality given the inherent complexity of reward modeling in large-scale systems. Additionally, we have provided empirical support for this claim through experiments with a toy model, which illustrate the challenges in achieving an "ideal" RM.
>
> **Q2: Questions about the theoretical base of the paper.**
>
> **R2:** In Appendix B, we showed in detail why in the proposed BRME, the std reflects the confidence of the head, and why it can be referred a measurement of the model uncertainty. To summarize, a positive gradient in MSE indicates that with more optimization steps, the output std $\sigma_i^+$ (or $\sigma_i^-$) decreases, reflecting higher confidence from the reward head in its scoring.
>
> More broadly, capturing the uncertainty set and conducting robustness trade-offs is well-established in traditional reinforcement learning, as discussed in Section 2.2. Our work extends the applicability of this theoretical framework to LLM alignment, bridging the gap between established concepts in traditional RL and practical implementation in large language models.
>
> **Q3: The situation in the toy model experiment is artificially created to make sure that reward model is hard to learn.**
>
> **R3:** ***We strongly suggest the reviewer to give more explanation about the claim. The setting of the toy model is first introduced by [1] and we just follow it without any artificially manipulation***.
>
> [1] Xu S, Fu W, Gao J, et al. Is dpo superior to ppo for llm alignment? a comprehensive study[J]. arXiv preprint arXiv:2404.10719, 2024.
>
> **Q4: Detailed explanation about the main experimental results**
>
> **R4:** Here we provide a point-by-point explanation.
>
> - *What are 800 steps of PPO?* The gradient descent on the actor model for the PPO optimization is conducted for 800 steps. We believe this is a commonly understood concept among industry professionals and thus does not require extensive explanation. In Line 353, we also clarified that the experience batch size in PPO is set to be 128.
>
> - *Where weren't standard training runs for fixed number of epochs done. Why are we comparing performances between epochs before the algorithm has seen all the training data?* As we discuss in Section 5.2.2, we want to present the advantage of the proposed method in both short run and long run. In the short run, incorporating robustness improves stability and early performance, while over the long run, it results in consistent performance gains compared to standard RLHF. This approach allows us to highlight both immediate and sustained benefits of reward-robust RLHF throughout the training process.
>
> **Q5: Intention of the proof in section 6.2**
>
> **R5:** The intention of the proof in Section 6.2 is to demonstrate that, under a constant reward setting, the model remains stable and its performance is acceptable due to the additional constraint imposed by KL divergence. In Section 6.2 (as well as Figure 5), it demonstrates that under conditions where the reward model is imperfect or unreliable, the use of a robustness measure, mitigates the detrimental effects and helps maintain stability. This approach allows the model to perform more consistently, even in cases where reward signals are weak or stochastic, thus bringing it closer to a constant reward setting and reducing the risk of model degradation.

---

> ### Comment · Reviewer_JCaz · 2024-11-25
>
> * PPO optimization is conducted for 800 steps. We believe this is a commonly understood concept among industry professionals and thus does not require extensive explanation.
>
> I am not sure this is an acceptable assumption. A paper should be rigorous in explaining its experimental protocol and not leave it on the assumption. Second, I don't think 800 steps of PPO is standard, and I think they are quite a few. Maybe the authors can cite the studies that do this for LLMs.
>
> * Bayesian intuition
>
> The authors added an explanation for why their method is Bayesian in Appendix B. I don't see that explanation as sound and grounded in principles but rather seems to be an intuitive explanation.
>
> * In Appendix B, we showed in detail why in the proposed BRME, the std reflects the confidence of the head, and why it can be referred a measurement of the model uncertainty
>
> I referred to appendix B but I still find the explanation of why BRME is argued to be doing the uncertainty measurement in a theoretically grounded way to be lacking. How does Decreasing std deviation with more gradient steps show it is indeed a true confidence?
>
>
>
> * Line 36: RLHF leading to the breakthrough of o1 models
> The grammar in the paper does imply this and seems to common confusion with another reviewer as well. But this is an minor issue.
>
> Thanks for clarifying the experimental setup. The toy setup seems okay to me now.
>
> * Comments
>
> I think the paper needs more work to be ready for publication in terms of principled claims, theory, and experiments.

---

### Official Review · Reviewer_3SmB · 2024-11-11

**Soundness:** 3
**Presentation:** 3
**Contribution:** 2
**Rating:** 5
**Confidence:** 3

**Summary:**

This paper presents an approach on building robust reward models (RMs) for stable RLHF. The main idea is to build a Bayesian reward model ensemble to model the uncertainty set of reward functions and then run RLHF with interpolation with the standard RLHF objective and the robust RLHF objective using the minimum reward score from the RM ensemble in the uncertainty set. The authors empirically show that the resulting approach can show gains in most of the language and reasoning tasks. The authors also show some theoretical insights and performed various ablations to show that the proposed method can indeed lead to underestimated reward scores and mitigate reward hacking.

**Strengths:**

1. The proposed method is simple yet effective. The idea is intuitive and easy to follow. The authors also provided some theoretical justifications on the algorithm, adding up the soundness of the paper.

2. The authors performed thorough evaluations on various benchmarks to show that the interpolation between standard RLHF and robust RLHF can lead to consistent gains across most of the benchmarks, making the paper more convincing.

3. The paper also includes some ablations to show that the proposed method indeed leads to underestimated rewards and can be useful for reducing reward hacking.

**Weaknesses:**

1. While the proposed method is neat, the difference between the method and previous approaches such as [1] and [2] is small IMO. The authors should discuss the comparisons in more details and also conduct some experiments to compare to [1] and [2] to show the superiority of the proposed algorithm. Without such discussion and empirical evidence, it's not clear if the paper is fully novel and improves over prior works.

2. While the authors show that the method can consistently improve over standard RLHF in various benchmarks, the gain seems relatively small (roughly within 1%-3%). Such small improvement could be just within the error bar of each of the benchmarks. The authors should also report the error bar of the results in the experiment section.

**Questions:**

see weakness section above.

---

> ### Author Response · Authors · 2024-11-20
>
> Thank you for your feedback; we will continue to improve our paper. Could you please provide more specific information about the papers referenced as [1] and [2]? It appears that the related references were not included in your comments.

---

### Meta-Review · Area_Chair_kHmh · 2024-12-22

**Metareview:**

The main idea is to build a Bayesian reward model ensemble to model the uncertainty set of reward functions and then run RLHF with interpolation with the standard RLHF objective and the robust RLHF objective using the minimum reward score from the RM ensemble in the uncertainty set.. Several concerns were raised in regards to the training procedure and more importantly comparison with prior work on reward model ensembles, which I also agree with. I don't think the authors' response is adequate in terms of comparisons to existing work in this area. I also think that the paper could better connect with other literature on robust RL to better present insights about the area. Finally, the theoretical statements are presented in ways that do not appear rigorous at places.

Unfortunately we are not able to accept the paper at this moment, but I would recommend authors to kindly look at the discussion and some of the comments above to prepare the revision of the paper.

**Additional Comments On Reviewer Discussion:**

Main points taken into account for the decision were (1) connection to prior work, (2) rigor of experiments and theory, and (3) terminology and concepts (e.g., overscoring vs underscoring is all about pessimism in RL / robust MDPs), and (4) unclear justification of some of the components of the approach (e.g., Bayesian ensembles vs standard ensembles).

---

### Decision · Program_Chairs · 2025-01-22

Reject